# Modulation of cardiomyocyte contractility and action potentials with chemogenetic chloride currents

Muhammed Sönmez[1,2] (ID), Tim Stüdemann[1,2], Christoph Manthey[1,2], Anita Covic[1], Nancy Shehata[1,2] (ID), Junsoo Im[1,2], Barbora Schwarzova[1] (ID), Judith Rössinger[1,2] (ID), Rajiven Srikantharajah[1,2], Aya Shibamiya[1,2], Edzard Schwedhelm[2,3], Thomas Eschenhagen[1,2], Torsten Christ[1,2] (ID) and Florian Weinberger[1,2,4] (ID)

[1]*Institute of Experimental Pharmacology and Toxicology, University Medical Center Hamburg-Eppendorf, Hamburg, Germany*

[2]*German Centre for Cardiovascular Research (DZHK), partner site Hamburg/Kiel/Lübeck, Hamburg, Germany*

[3]*Institute of Clinical Pharmacology, University Medical Center Hamburg-Eppendorf, Hamburg, Germany*

[4]*Research Group Tissue Engineering and Regenerative Therapies, Centro Nacional de Investigaciones Cardiovasculares Carlos III (CNIC), Madrid, Spain*

Handling Editors: Kim Barrett & Brian Delisle

The peer review history is available in the Supporting Information section of this article (https://doi.org/10.1113/JP286428#support-information-section).

**Abstract figure legend** Pluripotent stem cell derived-cardiomyocytes expressing the chemogenetic tools PSAM-GlyR and PSAM[4]-GlyR were used to generate engineered heart tissue. PSAM-GlyR and PSAM[4]-GlyR activation modulated cardiomyocyte action potentials and contractility. Created in BioRender.com.

**Abstract**   Transient perturbation of electrical activity is used in neuroscience to study the impact of specific neuronal cell populations on brain function. Similarly, cardiomyocyte (CM) physiology can be controlled by the activation of artificially expressed ion channels. Pharmacologically selective actuator modules (PSAMs) are engineered ligand-gated ion channels that can be activated with small molecules. We aimed to use the 'inhibitory' PSAMs, (i) PSAM$^{L141F,Y115F}$-GlyR (PSAM-GlyR) and (ii) PSAM$^{L131G,Q139L,Y217F}$ (ultrapotent PSAM[4]-GlyR), which consist of modified $\alpha$7-nicotinergic

M. Sönmez, T. Stüdemann, T. Christ and F. Weinberger contributed equally to this work.

acetylcholine receptor ligand binding domains and the ion pore domain of the glycine receptor, to modulate CM physiology with chloride currents. We employed CRISPR/Cas9 to integrate PSAM-GlyR and PSAM[4]-GlyR in induced pluripotent stem cells, differentiated CMs and generated engineered heart tissue (EHT). Video optical force recordings, sharp microelectrode action potential measurements and patch-clamp technique were used to characterize PSAM-GlyR and PSAM[4]-GlyR CMs. PSAM-GlyR and PSAM[4]-GlyR activation allowed titration of chloride currents in a reversible manner. We found that chloride currents modulated action potential characteristics. Patch clamp recordings showed that channel activation resulted in chloride-driven currents that depolarized the cell. In EHT, this resulted in a stop of contractility that was fully reversible after wash-out. We provide a comprehensive characterization of the chemogenetic tools PSAM-GlyR and PSAM[4]-GlyR in CMs, demonstrating their utility to modulate CM activity *in vitro* (PSAM-GlyR and PSAM[4]-GlyR) but also potential for *in vivo* applications (PSAM[4]-GlyR).

(Received 14 February 2024; accepted after revision 22 January 2025; first published online 22 February 2025)

**Corresponding authors** T. Christ and F. Weinberger: Institute of Experimental Pharmacology and Toxicology, University Medical Centre Hamburg-Eppendorf 20246, Germany.    Email: t.christ@uke.de, florananton.weinberger@cnic.es

**Key points**

- Pharmacologically selective actuator modules (PSAMs) are engineered ligand-gated ion channels that can be activated with small molecules.
- These chemogenetic tools have been applied in neuroscience to inhibit neuronal activity.
- Chemogenetic tools can also be used to modulate cardiomyocyte physiology.
- Activation of the PSAMs, PSAM-GlyR and PSAM[4]-GlyR depolarized cardiomyocytes and thus stopped cardiac contractility.
- Our study characterizes novel tools that can be used to modulate cardiomyocyte physiology in vitro and in vivo.

## Introduction

Conventional antiarrhythmic drugs modulate cardiac physiology via ion channels physiologically expressed in cardiomyocytes (CMs). Cardiac expression of non-cardiac ion channels has been proposed as a tool to fine-tune electrophysiological properties, e.g. with light (optogenetics) or small molecules (chemogenetics). In this study we used the pharmacologically selective actuator modules (PSAMs) PSAM[L141F,Y115F]-GlyR (PSAM-GlyR) and the ultrapotent PSAM PSAM[L131G,Q139L,Y217F] (PSAM[4]-GlyR) to modulate CM physiology and study the effect of increased chloride conductance on human pluripotent stem cell-derived CMs (hiPSC-CMs). PSAMs are engineered ligand-gated ion channels based on modified $\alpha$7-nicotinergic acetylcholine receptor ligand binding domains. Their ligand binding domains have been mutated to selectively interact with synthetic agonists (pharmacological selective effector molecules, PSEMs) (Magnus et al., 2011, 2019). For PSAM-GlyR and PSAM[4]-GlyR, the $\alpha$7-nicotinergic acetylcholine receptor ligand binding domains have been fused to the chloride-selective ion pore domain of the glycine receptor $\alpha$1-subunit. PSAM-GlyR and PSAM[4]-GlyR

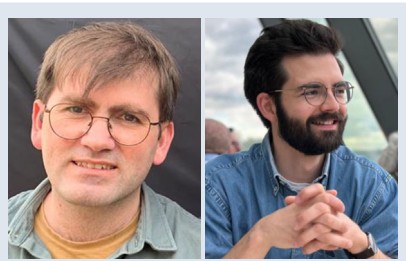

**Muhammed Sönmez** studied medicine at Istanbul University Cerrahpasa Medical Faculty, where he also obtained his PhD. Since 2021, he has been working as a postdoctoral researcher in the lab of Prof. Thomas Eschenhagen, under the supervision of PD. Dr. Torsten Christ, focusing on cardiac electrophysiology. **Tim Stüdemann** obtained his PhD in the lab of Prof. Thomas Eschenhagen, working on the mechanisms underlying cardiac cell therapy. He employed optogenetic and chemogenetic tools to understand cardiomyocyte physiology in engineered heart tissues and after transplantation in small animal models. Dr. Stüdemann is now working at TRON, Mainz (Germany), developing mRNA therapeutics for cardiovascular regeneration.

were described as inhibitory PSAMs and used to silence neuronal activity *in vitro* and *in vivo*, resulting from a substantially reduced input resistance (electrical shunting) (Magnus et al., 2011, 2019). Yet, a recent study described enhanced activity after the activation of $PSAM^4$-GlyR in dopaminergic medium spiny neurons (Gantz et al., 2021). In CMs chloride channels were identified in 1989, but their role is less well understood than the role of sodium and potassium channels. It has been particularly challenging to study effects of chloride channel activation because chloride channel blockers such as 4,4′-diisothiocyano-2,2′-stilbenedisulfonic acid, 9-anthracene carboxylic acid, tamoxifen and $Cd^{2+}$ lack specificity and genetic modifications are non-reversible and laborious (Varró et al., 2021).

At present only very few applications have been reported for chemogenetic tools in CMs (Stüdemann et al., 2022; Wexler et al., 2023). Designer receptors solely activated by designer drugs (DREADD) have been used to study Gq signalling in CMs (Kaiser et al., 2019). Recently, the engineered ligand-gated ion channel $PSAM^4$-5-HT3 was proposed as a tool for gene therapy to modulate cardiac electrophysiology. In CMs expressing $PSAM^4$-5-HT3, in which the modified nicotinergic ligand binding domain was fused with the 5-HT3 receptor ion pore domain, activating the construct with 10 nM varenicline depolarized the membrane potential to −10 mV and stopped beating in isolated CMs. In contrast, in a cell sheet tissue model, varenicline concentrations between 5 and 30 nM increased beating frequency and only a higher varenicline concentration of 100 nM slowed, and even completely stopped, spontaneous activity. The authors suspected diffusion issues of varenicline to be responsible (Wexler et al., 2023). Interpretation is complicated since the conductivity of artificial channels can include several different ions, and the relationship between effects on membrane potential and beating rate may not be linear. Therefore, we have compared effect sizes of drug-induced currents and subsequent changes in excitability with the chloride-selective engineered ion channels PSAM-GlyR and $PSAM^4$-GlyR. PSAM-GlyR can be activated only with compounds not approved for use in humans, but the slightly modified variant $PSAM^4$-GlyR can be activated by the partial nicotinergic receptor agonist varenicline, approved as a smoking cessation drug (Magnus et al., 2019). Patch clamp recordings demonstrated that channel activation resulted in chloride-driven currents that depolarized the cell. We found that chloride currents could be titrated in a reversible manner. In engineered heart tissue (EHT), this resulted in a stop of contractility that was fully reversible after wash-out. In summary, we provide a comprehensive characterization of the novel chemogenetic tools PSAM-GlyR and $PSAM^4$-GlyR in CMs and describe their utility to 'inhibit' CM activity.

## Methods

### Engineering of the PSAM-GlyR and $PSAM^4$-GlyR hiPSC-lines

We used the previously established iPSC line UKE-001A (https://hpscreg.eu/cell-line/UKEi001-A) with an intact karyotype at p26 as control cell line and as parental cell line for gene editing. The generation of the PSAM-GlyR cell line was recently described (Stüdemann et al., 2022). PSAM-GlyR and $PSAM^4$-GlyR were generated by targeted knock-in. AAVS1-CAG-hrGFP was a gift from Su-Chun Zhang (Addgene plasmid no. 52344). CAG::PSAML141F, Y115F:GlyR-IRES-GFP and pCAG $PSAM^4$ GlyR IRES EGFP were gifts from Scott Sternson (Addgene plasmid no. 32480 and Addgene plasmid no. 119739). The plasmid vectors contained PSAM-GlyR or $PSAM^4$-GlyR under the control of a CAG promoter, homology arms (each ∼800 bp in size) for the *AAVS1* locus, and a puromycin resistance cassette upstream of the promoter. PSAM-GlyR and $PSAM^4$-GlyR were linked to enhanced green fluorescent protein (EGFP) by a 2A linker. Nucleofection with the Cas9 ribonucleoprotein was conducted using a 4D-Nucleofector (Lonza, Basel, Switzerland) according to the manufacturer's protocol. The selected single guide RNA targeted the *AAVS1* locus between Exon 1 and 2 and was designed using CRISPOR (http://crispor.tefor.net). Positively edited cells were enriched by fluorescence-activated cell sorting (Aria Fusion, BD Biosciences, San Jose, CA, USA) and seeded as single cells. Correct transgene integration was verified by PCR amplification followed by Sanger sequencing. The clone characterization strategy was adapted from Oceguera-Yanez et al. (2016). For each clone three PCR reactions were performed. One to confirm correct 5′ insertion, one to confirm correct 3′ insertion and one to detect the unedited allele. This combination of reactions allowed us to discriminate homozygous from heterozygous clones. We then performed PCR for the whole transgene including the insertion sites from a subset of clones. These PCR fragments were subsequently Sanger sequenced. Primer sequences are listed in Table 1. An overview of this strategy can be found in A1. Homozygous clones were used for this study (clone ASC3 for $PSAM^4$-GlyR; A1). Karyotype was assessed using the Nanostring Karyotype panel by the Nanostring Core Facility of the University Medical Center Hamburg-Eppendorf (UKE) and did not show major genomic aberrations (A1).

**Table 1. Primer sequences for genotyping**

| Primer Name | Primer colour in illustration A3 | Sequence |
|---|---|---|
| 3′ insert forward | Yellow | CTGGGGTGCCTAATGAGTGA |
| CAG for | Pink | GCTCTAGAGCCTCTGCTAACCATG |
| EYFP rev | Blue | CGGACACGCTGAACTTGTGG |
| HA-L genomic for | Red | TCGACCTACTCTCTTCCGCA |
| HA-L insert rev | Green | GTACTCGGTCATCTCGAGCC |
| HA-R genomic rev | Black | ACAGTGACCAACCATCCCTG |
| HA-R insert rev | Red | GGCTCCATCGTAAGCAAACC |

## EHT from hiPSC-CMs expressing PSAM-GlyR or PSAM$^4$-GlyR

EHTs were generated from cells and fibrinogen/thrombin as previously described (Breckwoldt et al., 2017). Control, PSAM-GlyR or PSAM$^4$-GlyR-CMs were digested with collagenase II (Worthington, NJ, USA, LS004176; 200 U/mL) in $Ca^{2+}$-free Hanks' Balanced Salt Solution (HBSS) (Thermo Fisher Scientific, Waltham, MA, USA, 14175-053) with 1 mM HEPES (pH 7.4), 10 μM Y-27632, and 30 μM *N*-benzyl-*p*-toluene sulphonamide (TCI, Eschborn, Germany, B3082) for 3.5 h at 37°C (5% $CO_2$, 21% $O_2$). The dissociated cells were resuspended in $Ca^{2+}$-containing Dulbecco's modified Eagle's medium (DMEM) with 1% penicillin/streptomycin. Cell concentration was adjusted to $10–15 \times 10^6$ cells/mL. Fibrin-based human EHTs were generated in agarose casting moulds with solid silicone racks (100 μL per EHT, $1 \times 10^6$ cells). The culture medium was changed three times per week. Analysis of contractile force was performed by video-optical recording. The contraction peak analysis was performed during spontaneous beating. EHTs were measured based on automated video-optical recording. In short, a video camera was placed above the recording chamber, the deflection of the silicone posts was recorded and analysed with specific software (CTMV, Pforzheim, Germany). The contraction peaks were analysed in terms of frequency, force and contraction (time-to-peak, TTP) and relaxation time (RT80) at 80% of peak (Mannhardt et al., 2016).

## Drugs, solutions and concentrations

The compounds varenicline (PZ0004) and picrotoxin (P1675) were provided by Sigma-Aldrich (St Louis, MO, USA), while the pharmacologically selective effector molecule 89S (PSEM$^{89S}$, cat. no. 6426), was obtained from Tocris (Bristol, UK). Varenicline and PSEM$^{89S}$ were prepared as stock solutions in phosphate buffered saline (PBS). Picrotoxin (PTX) was dissolved in dimethyl sulfoxide. The solutions were then aliquoted and stored at −20°C. On the day of the experiment, the stock solutions were warmed to room temperature. Varenicline was diluted to concentrations ranging from 1 nM to 1 μM, PSEM$^{89S}$ from 1 μM to 1 mM and PTX was diluted to 50 mM. Dilutions were prepared using the bath solution mentioned above for patch clamping, Tyrode's solution for action potential recordings and $Ca^{2+}$-containing DMEM for force measurements.

## Current measurements

Ion currents were measured at 37°C in the whole-cell configuration using an Axopatch 200B amplifier (Axon Instruments, Foster City, CA, USA). ISO2 software (MFK, Niedernhausen, Germany) was used for data acquisition and analysis. Heat-polished pipettes were pulled from borosilicate filamented glass with an external diameter of 1.5 mm and internal diameter of 0.87 mm (HILG1103227; Hilgenberg, Malsfeld, Germany) with a DPZ-Universal puller (Zeitz Instruments, Munich, Germany). Tip resistances were 2.5–5 MΩ, seal resistances were 3–6 GΩ. Cell capacitance ($C_m$) was calculated from steady-state current during depolarizing ramp pulses (1 V/s) from −40 to −35 mV. The cells were investigated in a small perfusion chamber placed on the stage of an inverted microscope. Inward currents were measured with the following bath solution (in mM): NaCl 120, KCl 5.4 and HEPES 10, $CaCl_2$ 2, $MgCl_2$ 1 and glucose 10 (pH 7.4, adjusted with NaOH). The internal solution included (in mM): DL-aspartate potassium salt 80, KCl 40, NaCl 8, HEPES 10, Mg-ATP 5, Tris-GTP 0.1, EGTA 5 and $CaCl_2$ 2 (pH 7.4, adjusted with KOH). Currents were measured at a potential of −75 mV, if not otherwise indicated. Cells were exposed to increasing concentration of PSEM$^{89S}$ and varenicline.

Voltage-dependency of inward current activation was estimated following different test pulse potentials (from −100 to −20 mV, increment 10 mV). We calculated voltage errors in our settings by the following equation (Grey & Santin, 2023):

$$\text{Effective resistance } (R_{\text{eff}}, \text{ M}\Omega) = \text{Seal resistance } (R_{\text{s}})$$
$$\times (1 - \%\text{Comp}/100)$$

$$\text{Voltage error} = V_{\text{Pipette}} - V_{\text{Membrane}} = I_{\text{f}} \text{ (feedback current)}$$
$$\times R_{\text{eff}}$$

We measured an $R_{\text{s}}$ of $4.9 \pm 0.1$ M$\Omega$ ($n = 134$), which was compensated by 80% giving a $R_{\text{eff}}$ of 0.98 (M$\Omega$). Even with larger Cl$^-$ current amplitudes, resulting voltage error was not larger than 1 mV.

Concentration–response curves and reversal linear regression fit to estimate the reversal potential were plotted with GraphPad Prism software version 6 (GraphPad Software, San Diego, CA, USA).

### Action potential measurement

Actions potential (AP) measurements in PSAM-GlyR and PSAM$^4$-GlyR EHTs were performed with standard sharp microelectrodes pulled from the glass capillaries mentioned above (Lemoine et al., 2017). Tip resistance was between 25 and 50 M$\Omega$ when filled with 2 M KCl. The EHTs were transferred from the 24-well EHT culture plate into the AP measuring chamber by cutting the silicone posts, and were fixed with needles in an optimal position for AP recording. Tissues were continuously superfused with Tyrode's solution (in mM: NaCl 127, KCl 5.4, MgCl$_2$ 1.05, CaCl$_2$ 1.8, glucose 10, NaHCO$_3$ 22, NaHPO$_4$ 0.42, balanced with 95% O$_2$–5% CO$_2$ at 36°C, pH 7.4).

The signals were amplified by a BA-1s npi amplifier (npi electronic GmbH, Tamm, Germany). APs were recorded and analysed offline using Lab-Chart software (version 8, ADInstruments Pty Ltd, Castle Hill, NSW, Australia). Drug effects were measured 15 min after adding the substances. AP parameters were averaged from 15 consecutive APs.

## Results

### PSAM-GlyR and PSAM$^4$-GlyR EHTs showed subtle physiological alterations

We differentiated CMs from hiPSCs expressing PSAM-GlyR or PSAM$^4$-GlyR. The unedited parental hiPSC line (UKEi001-A) served as control. Both edited cell lines retained pluripotency (stage-specific embryonic antigen 3 (SSEA3) positivity >99%) and could be differentiated to CMs with high purity (cardiac troponin-t (cTnt) positivity >90%). Transgene expression (based on GFP positivity) was $93 \pm 7\%$ in hiPSC-CMs (Fig. 1A and B). We then generated EHTs to study the physiological consequences of PSAM-GlyR or PSAM$^4$-GlyR expression and the impact of channel activation on electrical activity and contractility.

Similar to WT EHT, PSAM-GlyR and PSAM$^4$-GlyR EHTs coherently started to beat after 1 week of culture. CMs of both cell lines matured in the EHT format. $\alpha$-Bungarotoxin toxin was used to label PSAM-GlyR in EHT-CMs. We cannot exclude that individual cells did not express the transgene, but $\alpha$-bungarotoxin staining demonstrated transgene expression at least in the vast majority of CMs (Fig. 1C). On average contractile force of PSAM-GlyR and PSAM$^4$-GlyR EHTs did not differ from control EHT after 4 weeks of culture (the time point when EHTs were used for further experiments; control: $0.18 \pm 0.02$ mN; PSAM-GlyR: $0.17 \pm 0.04$ mN and PSAM$^4$-GlyR: $0.16 \pm 0.06$ mN; $P = 0.7865$ for PSAM-GlyR vs. control and $P = 0.3196$ for PSAM$^4$-GlyR vs. control, $n = 19/39/31$ EHTs from $n = 1/3/3$ batches; Fig. 1D), but there was greater variability in EHTs from genetically modified CMs compared to WT EHT. Beating frequency was similar in PSAM-GlyR EHT and slightly higher in PSAM$^4$-GlyR compared to controls (control: $48.6 \pm 7.3$ bpm, PSAM-GlyR: $49.1 \pm 18.4$ bpm and PSAM$^4$-GlyR: $57.7 \pm 20.7$ bpm; $P = 0.9955$ for PSAM-GlyR vs. control and $P = 0.1844$ for PSAM$^4$-GlyR vs. control, $n = 19$ (control)/39 (PSAM-GlyR)/31 (PSAM$^4$-GlyR) EHTs from $n = 1/3/3$ batches; Fig. 1E). Detailed physiological characterization revealed slightly longer relaxation time in PSAM$^4$-GlyR EHT (control: $0.20 \pm 0.02$ ms and PSAM$^4$-GlyR: $0.25 \pm 0.04$ ms; $P = 0.0112$; but not TTP), while TTP and relaxation time did not differ between control and PSAM-GlyR EHT (TTP: control: $0.15 \pm 0.01$ ms and PSAM-GlyR: $0.15 \pm 0.01$ ms; $P = 0.8606$; relaxation time: control: $0.20 \pm 0.02$ ms and PSAM-GlyR: $0.20 \pm 0.06$ ms; $P = 0.9990$, $n = 19$ (control)/39 (PSAM-GlyR)/31 (PSAM$^4$-GlyR) EHTs from $n = 1/3/3$ batches; Fig. 1F and G).

### Depolarized diastolic potential in EHT from PSAM-GlyR but not from PSAM$^4$-GlyR

Given the small differences in contractility and frequency, we assessed whether the expression of PSAM-GlyR and PSAM$^4$-GlyR altered electrical activity of EHT. For this purpose, we measured AP in PSAM-GlyR and PSAM$^4$-GlyR EHT in the absence of an agonist and compared data to the control line. Maximum diastolic potential (MDP) was close to $-80$ mV and action potential duration at 90% repolarization (APD$_{90}$) accounted for about 240 ms in control EHT (Fig. 2A and E). While AP trace and parameters in PSAM$^4$-GlyR EHT were similar to control EHT (Fig. 2A–F), PSAM-GlyR EHT revealed electrophysiological differences. MDP was less negative ($68.2 \pm 6.8$ vs. $78.2 \pm 2.8$ mV, $P < 0.0001$ for PSAM-GlyR vs. control and $68.2 \pm 6.8$ vs. $76.3 \pm 4.9$ mV, $P = 0.0014$ for PSAM-GlyR vs. PSAM$^4$-GlyR), the AP amplitude (APA) was smaller ($95.6 \pm 11.2$ vs. $117.1 \pm 9.2$ mV, $P < 0.0001$ for PSAM-GlyR vs. control

and $95.6 \pm 11.2$ *vs.* $109.3 \pm 12.1$ mV, $P = 0.0172$ for PSAM-GlyR *vs.* PSAM$^4$-GlyR) and maximum upstroke velocity ($V_{max}$) was slower in PSAM-GlyR than in control EHT ($104.4 \pm 78.6$ *vs.* $175.8 \pm 60.9$ V/s, $P = 0.0437$ for PSAM-GlyR *vs.* control and $104.4 \pm 78.6$ *vs.* $191.4 \pm 86.2$ V/s, $P = 0.0232$ for PSAM-GlyR *vs.* PSAM$^4$-GlyR; $n = 16/3$ (Control), $10/3$ (PSAM) and $11/3$ (PSAM$^4$) EHTs/batches Fig. 2*B*, *C* and *F*).

## PSAM-GlyR expression in hiPSC-CMs was associated with a constitutively active current

The partial depolarization (i.e. less negative diastolic potential) in PSAM-GlyR EHT compared to control and PSAM$^4$-GlyR EHT suggested some constitutive currents in PSAM-GlyR-expressing hiPSC-CMs (Fig. 3*A*). To study this hypothesis, we compared currents measured at $-75$ mV in hiPSC-CMs expressing PSAM-GlyR and PSAM$^4$-GlyR hiPSC-CMs in the absence of the agonists PSEM$^{89S}$ (PSAM-GlyR) and varenicline (PSAM$^4$-GlyR), respectively. Currents at $-75$ mV were larger in PSAM-GlyR- than in PSAM$^4$-GlyR-expressing

hiPSC-CMs ($352.8 \pm 289.6$ *vs.* $87.0 \pm 88.9$ pA, $P < 0.0001$, Fig. 3; current density: $6.2 \pm 5.8$ *vs.* $2.77 \pm 2.8$ pA/pF, $P = 0.0013$, mean $\pm$ SD, $n = 17/5/3$ (PSAM) and $n = 20/5/3$ (PSAM$^4$), cells/EHTs/batches, Fig. 4). To determine the contribution of PSAM-GlyR to the current, we exposed cells to PTX (Sigma-Aldrich, P1675), recently reported to block varenicline-induced currents in PSAM$^4$-5-HT3-expressing hiPSC-CMs (Wexler et al., 2023). As expected, PTX (1 mM) had no effect on currents at $-75$ mV in control hiPSC-CMs, but it also did not exert an effect in PSAM-GlyR and PSAM$^4$-GlyR-hiPSC-CMs. Similarly, pre-treatment with PTX did not prevent or reduce PSEM$^{89S}$- ($49.9 \pm 17.1$ *vs.* $51.9 \pm 11.3$, $P = 0.8637$) or varenicline-induced currents ($36.4 \pm 28.6$ *vs.* $39.8 \pm 20.3$ pA/pF, $P = 0.7220$; Fig. 3*B* and *C*, $n = 4/2/1$ (PSAM) and $17/3/2$ (PSAM$^4$), cells/EHTs/batches).

## PSAM-GlyR and PSAM$^4$-GlyR activation generated chloride currents

PTX was reported to partially block the chloride channel pore of PSAM$^4$-GlyR (Magnus et al., 2019). From the

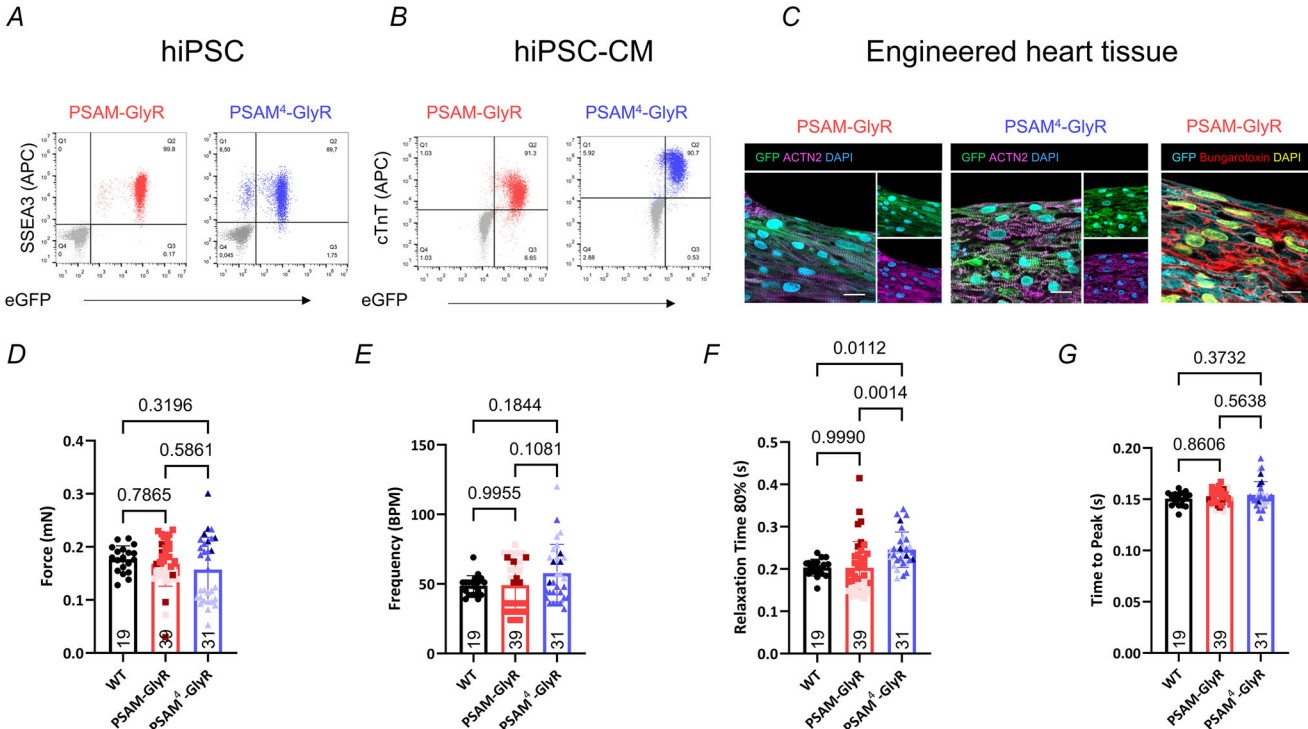

**Figure 1. Baseline characterization of PSAM-GlyR and PSAM$^4$-GlyR hiPSC and hiPSC-cardiomyocytes**
*A* and *B*, flow cytometry of either PSAM-GlyR or PSAM$^4$-GlyR human induced pluripotent stem cells (hiPSCs; *A*) or hiPSC-CMs (*B*) stained for SSEA3 or cardiac troponin T. *C*, longitudinal sections of a PSAM-GlyR and PSAM$^4$-GlyR EHT stained for GFP and ACTN2. $\alpha$-Bungarotoxin was used to label PSAM-GlyR. *D–G*, video-optical contractility measurement of control, PSAM-GlyR or PSAM-GlyR$^4$ EHT ($n = 19/39/31$ EHTs from $n = 1/3/3$ batches). *D*, contractile force at day 28 of culture. *E*, beating frequency at day 28 of culture. *F*, relaxation time at day 28 of culture. *G*, contraction time at day 28 of culture. Groups were compared using one-way ANOVA with multiple comparison, and *P*-values are provided in the graph. Measurements were performed in culture medium. Plotted are means $\pm$ SD. [Colour figure can be viewed at wileyonlinelibrary.com]

failure of PTX to block PSEM[89S]- and varenicline-induced inward currents in our experiments one might suspect that the PSAM-GlyR- and PSAM[4]-GlyR-induced currents in hiPSC-CMs were not chloride-driven. Therefore, we studied whether the charge carrier for PSAM-GlyR- and PSAM[4]-GlyR-induced currents was in fact chloride. We measured PSAM-GlyR- and PSAM[4]-GlyR-induced currents at a wide range of test pulse potentials. $I–V$ curves were plotted to calculate the reversal potential of the PSEM[89S] (PSAM-GlyR)- and varenicline (PSAM[4]-GlyR)-induced currents. When corrected for the calculated liquid potential, the reversal potential amounted to $-23$ mV, close to the calculated reversal potential of chloride current of $-25$ mV under our experimental conditions (chloride current concentration in the pipette solution: 52 mM and bath solution: 131 mM) indicating that activation of both engineered receptors indeed caused chloride currents (Fig. 4*A*, $n = 3/3/2$ (PSAM) and 4/4/2 (PSAM[4]) cells/EHTs/ batches).

### Varenicline potency was higher for PSAM[4]-GlyR than PSEM[89S] potency for PSAM-GlyR

Whereas PSAM-GlyR can be activated only with compounds not approved for use in humans in high (micromolar) concentration, the slightly modified variant PSAM[4]-GlyR was specifically designed for activation with clinically approved drugs in nanomolar concentrations allowing for an *in vivo* application. In fact, in neurons varenicline activated PSAM[4]-GlyR in nanomolar concentrations, whereas micromolar PSEM[89S] concentrations were necessary to activate PSAM-GlyR. We studied if this holds true also for CMs. For this purpose, we cumulatively exposed CMs to increasing concentrations of PSEM[89S] (PSAM-GlyR) or varenicline (PSAM[4]-GlyR, Fig. 4*B*). As expected, increasing drug concentrations resulted in larger currents. Transient spike-like increases in current made precise calculation of $E_{max}$ difficult (Fig. 4*B*). However, even with this uncertainty in mind, the EC[50] value of varenicline for

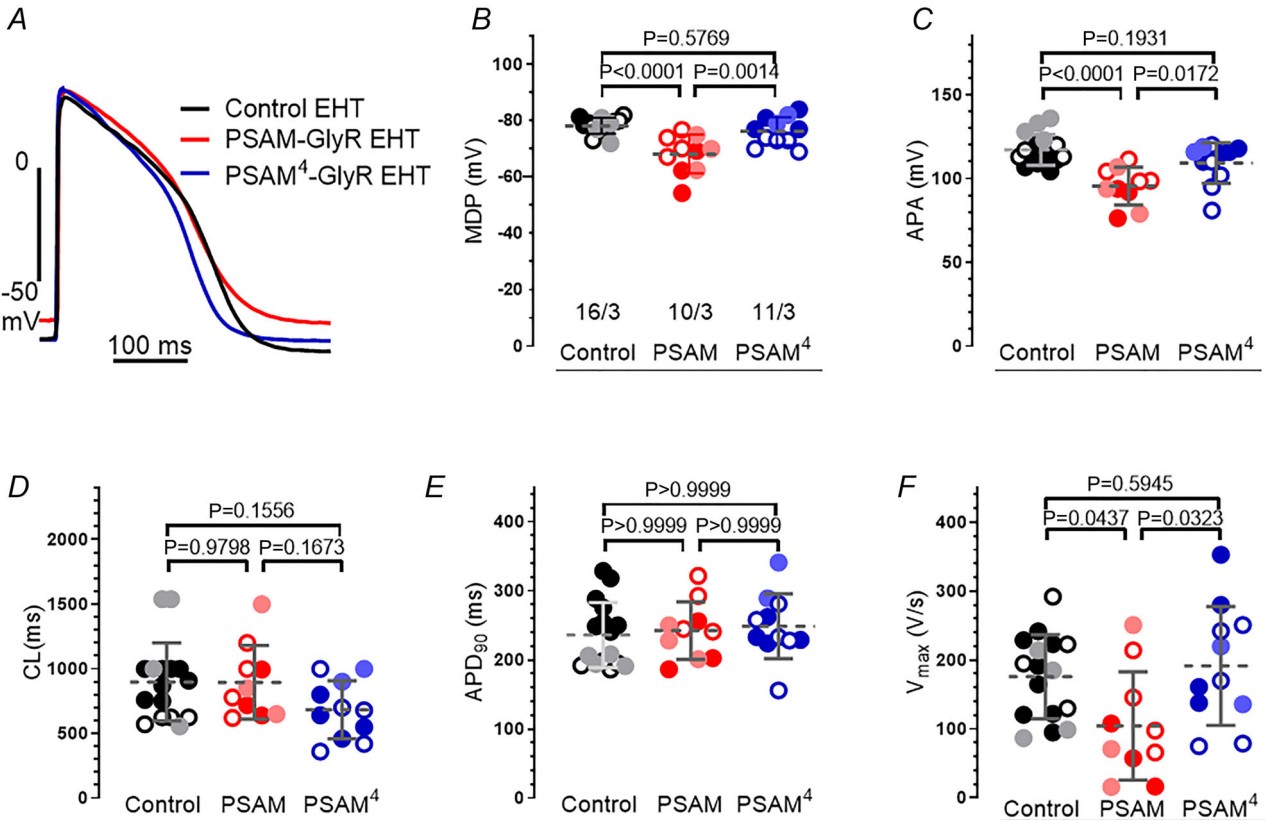

**Figure 2. PSAM-GlyR EHT but not PSAM[4]-GlyR EHT had a less negative maximum diastolic potential**
*A*, action potential traces recorded in control, PSAM-GlyR and PSAM[4]-GlyR EHT. *B–F*, mean values $\pm$ SD for various electrophysiological parameters. *B*, maximum diastolic potential (MDP); *C*, amplitude of action potential (APA); *D*, cycle length (CL); *E*, action potential duration at that 90% repolarization (APD[90]); *F*, maximum upstroke velocity ($V_{max}$). The number of EHT/number of batches is given as *n/n* and different shades indicate different batches. Groups (Control, PSAM: PSAM-GlyR and PSAM[4]: PSAM[4]-GlyR) are compared using one-way ANOVA, and *P*-values are provided in the graph. Measurements were performed in Tyrode's solution. [Colour figure can be viewed at wileyonlinelibrary.com]

PSAM[4]-GlyR was estimated to be a thousand-fold smaller than PSEM[89S] for PSAM-GlyR (~30 nM *vs.* ~10 μM, Fig. 4*C*, $n = 17/5/3$ (PSAM) and 20/5/3 (PSAM[4]), cells/EHTs/batches) and close to the originally reported values (Magnus et al., 2011, 2019).

### PSEM[89S] prolonged APD in control EHT

When assessing the effects of PSEM[89S] on EHT contractility, we noticed an increase in relaxation time with increasing concentrations. Therefore, we tested whether the agonists PSEM[89S] and varenicline may have off-target effects on other cardiac ion channels besides PSAM-GlyR and PSAM[4]-GlyR. For this purpose, we used control EHT. While varenicline (at concentrations slightly higher than the EC[50] for current activation (up to 50 nM)) had no effect on AP characteristics, equipotent concentrations of PSEM[89S] (at concentrations required for PSAM-GlyR activation (10–100 μM)) prolonged APD at voltages below 0 mV (A2, $n = 8/2$ EHTs/batches), suggesting relevant impairment of repolarizing forces.

### PSAM[4]-GlyR-induced currents depolarized diastolic potential in EHT

Activation of PSAM-GlyR and PSAM[4]-GlyR shifted the diastolic potentials in EHT to less negative values in a concentration-dependent manner (Fig. 5*A* and *B*, data for PSAM are given in A3, $n = 10/3$ EHTs/batches).

Maximum depolarization amounted to −30 and −35 mV. AP stopped after increasing varenicline concentration to 30 nM (indicated by the arrow) depending on a depolarization block and we could only measure MDP after this point (Fig. 5*B*). The EC[50] values for PSAM-GlyR- and PSAM[4]-GlyR-induced depolarization (measured by sharp microelectrode) were found close to EC[50] values measured in patch clamp experiments, indicating that full activation of PSAM-GlyR- or PSAM[4]-GlyR-induced currents is necessary to depolarize EHT close to the reversal potential of chloride (Fig. 5*C*). The drop in MDP was paralleled by a substantial decrease in $V_{max}$ (Fig. 5*C*, $n = 11/3$ EHTs/batches).

### Impact of PSAM[4]-GlyR-induced depolarization on beating rate in EHT

Next, we projected the size of the chloride currents determined by patch-clamp experiments on the AP shape in spontaneously beating EHT (Fig. 6*A* and *C*). For this purpose, we compared inward currents and AP at matching concentrations of varenicline. With inward currents up to 14 pA/pF (induced by 1–10 nM varenicline), the spontaneous beating rate increased gradually from ~65 bpm to 85 bpm. In parallel, APD[90] shortened gradually at all repolarization levels ($248.5 \pm 46.8$ *vs.* $214.0 \pm 58.7$, mean $\pm$ SD). However, only slightly larger currents (18 instead of 14 pA/pF)

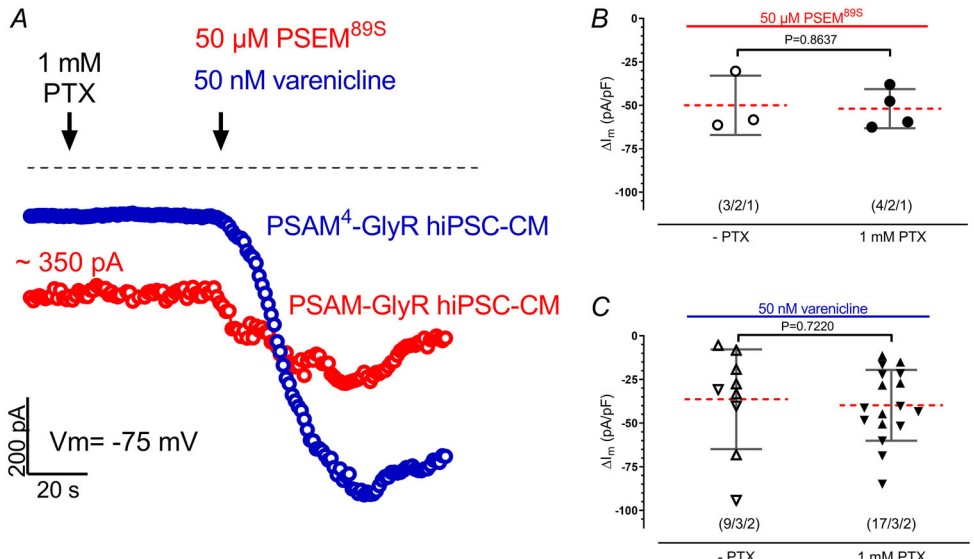

**Figure 3. PSAM-GlyR expression in hiPSC-CMs associated with some 'constitutively leak' current**
*A*, time courses of currents measured at −75 mV in hiPSC-CMs expressing PSAM-GlyR (red) and hiPSC-CMs expressing PSAM[4]-GlyR (blue). Cells were exposed to 1 mM picrotoxin (PTX) initially and then to 50 nM varenicline (hiPSC-CMs expressing PSAM[4]-GlyR) or 50 μM PSEM[89S] (hiPSC-CMs expressing PSAM-GlyR). *B* and *C*, mean values $\pm$ SD for increase in current by 50 μM PSEM[89S] or 50 nM varenicline in the absence and presence of 1 mM PTX. The number of hiPSC-CMs/number of EHTs/number of batches are given as *n/n/n* and different shapes within the groups indicate different batches. Groups are compared using unpaired *t* test, and *P*-values are provided in the graph. [Colour figure can be viewed at wileyonlinelibrary.com]

decreased beating rate drastically (6 out of 11 EHTs even stopped, Fig. 6B, $n = 11/3$ EHTs/batches), suggesting a depolarization block.

### Application of PSEM[89S] or varenicline stopped contractility in PSAM-GlyR and PSAM[4]-GlyR EHT, respectively

Next, we studied the effects of PSEM[89S] (PSAM-GlyR) and varenicline (PSAM[4]-GlyR) application on EHT contractility. As expected, PSEM[89S] and varenicline (PSAM[4]-GlyR) stopped contractility in a concentration-dependent manner in PSAM-GlyR or PSAM[4]-GlyR EHTs, respectively (Fig. 7A and B, $n = 7/11/8$ EHTs). EHT length measurement revealed that EHTs were stopped in diastole, in a fully relaxed state (Fig. 7C, $n = 4/3$ EHTs per group). Similar to the electrophysiological data, 100 μM PSEM[89S] was needed to fully inhibit contractility, while varenicline

inhibited contractility at a much lower concentration (50 nM). In accordance with the current measurements in isolated CMs, PTX did not inhibit the PSEM[89S]- or varenicline-induced stop of contractility. Contractility resumed quickly after wash-out of PSEM[89S] or varenicline. Importantly and somewhat surprisingly, EHTs could be pharmacologically stopped for days and even weeks without signs of toxicity and quickly recovered to the same force values as their time-matched controls (Fig. 7D and E, $n = 4/3$ EHTs per group). As iPSC-CMs mature over time in the EHT format, we went on to assess the functionality of the system after a cultivation period of 2 months. In accordance with the data for younger EHTs, PSEM[89S] stopped the contractility ($n = 3$ EHTs). This effect was again fully reversible (Fig. 7F).

### Discussion

Modulation of CM physiology offers a vast number of application possibilities in basic and translational

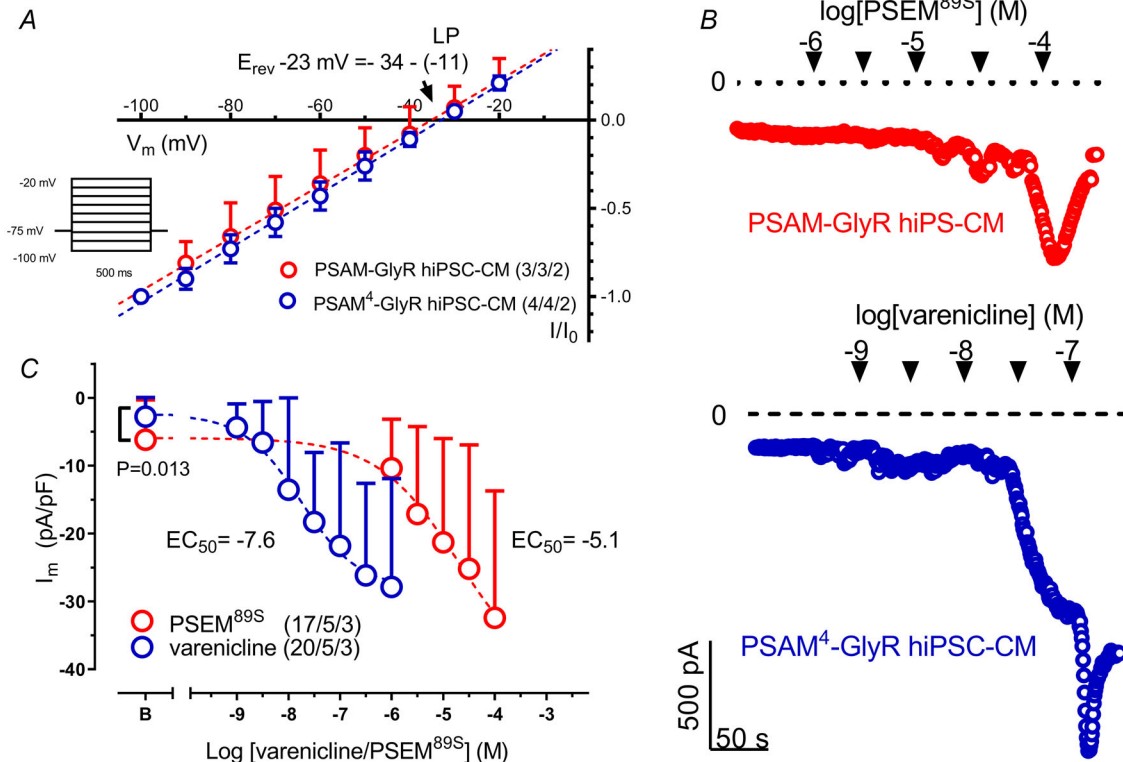

**Figure 4. Charge carrier for PSAM-GlyR and PSAM[4]-GlyR was chloride and the agonists PSEM[89S] and varenicline showed the expected potency**
*A*, current–voltage relationship of PSAM[89S]- and varenicline-evoked currents. The reversal potential and liquid–liquid junction potential (LLJP) are indicated by an arrow. Mean values ± SD for normalized currents. *n/n* indicates the number of hiPSC-CMs/number of EHTs. *B*, time courses of inward currents at −75 mV in hiPSC-CMs expressing PSAM-GlyR or PSAM[4]-GlyR exposed to increasing concentrations of PSEM[89S] or varenicline. *C*, concentration–effect curve for PSEM[89S] and varenicline. Basal current density of PSAM and PSAM[4]-GlyR-hiPSC-CMs is compared using unpaired *t* test, and *P*-values are provided in the graph. Mean values ± SD for chloride current density ($I_m$), and *n/n/n* indicates the number of hiPSC-CMs/number of EHTs/number of batches. [Colour figure can be viewed at wileyonlinelibrary.com]

cardiovascular science. CM physiology can be controlled with the activation of artificially expressed ion channels. Activation can be achieved with either light (optogenetics) or small (chemogenetics) molecules. Application of light within complex organ systems (e.g. the heart) represents a challenge. Therefore, modulation of cell physiology by chemogenetics has gained interest. We applied the chemogenetic ion channels PSAM-GlyR and PSAM[4]-GlyR in hiPSC-CMs to (i) modulate CM physiology and (ii) study the effect of titrated chloride channel activation on CM. For this we integrated the PSAM-GlyR and PSAM[4]-GlyR in the *AAVS1* locus of hiPSC and used clonally expanded cell lines. We have chosen a knock-in approach over a viral delivery to achieve a stable transgene expression (per cell but also in the cell population). Transgene expression was >90% (indicated by GFP positivity) in iPSC-CMs and remained stable over the whole study period. Repeated genotyping provided no evidence for clonal heterogeneity. The incomplete transgene expression might have resulted from random silencing in a small subpopulation (Bhagwan et al., 2020). Membranous integration was confirmed by $\alpha$-bungarotoxin staining that, even though it did not allow for a quantitative analysis, stained the vast majority of all CMs in the EHTs. The main findings of our study were: (i) activation of chloride channels in hiPSC-CMs generated currents large enough to clamp the membrane potential to the reversal potential of chloride; (ii) while moderate activation of chloride currents gradually increased the beating frequency of hiPSC-CMs, full activation inhibited beating abruptly; (iii) inhibition of cardiac contractility was fully reversible even after several weeks; and unexpectedly (iv) PTX did not block PSAM[4]-GlyR chloride currents; and (v) PSEM[89S] showed relevant off-target effects.

## PSAM-GlyR and PSAM[4]-GlyR activate CMs

PSAMs were initially developed to modulate neuronal activity. 'Inhibitory' PSAMs, such as PSAM-GlyR and PSAM[4]-GlyR, were specifically designed to silence neuronal activity, by electrical shunting. While this strategy has been successful in hippocampal pyramidal cells, PSAM[4]-GlyR activation resulted in increased activity in dopamine D1 receptor-expressing medium spiny neurons (D1-MSNs) in the ventral striatum. In spiny neurons, PSAM-GlyR[4] activation depolarized the membrane potential, which caused a depolarization block

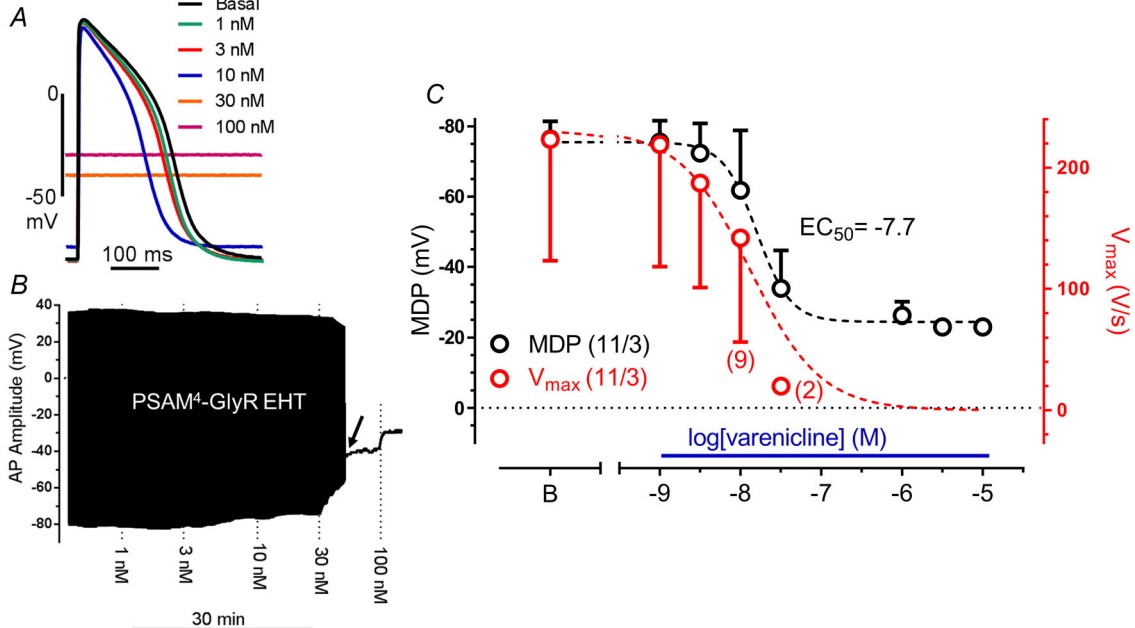

**Figure 5. Concentration-dependent effects of varenicline on action potential characteristics in PSAM[4]-GlyR EHT**

*A*, action potential traces at baseline and at increasing varenicline concentrations. *B*, time course of action membrane potential recorded in PSAM[4]-GlyR EHT exposed to increasing concentration of varenicline in PSAM[4]-GlyR EHT. AP stopped at a varenicline concentration of 30 nM (indicated by an arrow). The black line after this point represents only maximal diastolic potential (MDP). *C*, concentration–effect curve for varenicline effect on MDP and maximum upstroke velocity ($V_{max}$). Note that plotted EC$_{50}$ values are calculated from the MDP concentration–response curve. Two out of 11 EHTs stopped beating at 10 nM varenicline and 6 out of 11 EHTs stopped beating at 30 nM varenicline. Mean values $\pm$ SD for MDP (black) or $V_{max}$ (red). The number of EHTs/number of batches are given as *n/n*. [Colour figure can be viewed at wileyonlinelibrary.com]

in some cells, but most neurons showed subthreshold depolarization and increased AP firing (Gantz et al., 2021). In CMs, the maximal diastolic potential lies between −70 and −80 mV (Lemoine et al., 2018). The reversal potential for chloride in CMs is ∼−30 mV (Laflamme & Becker, 1996). Activation of chloride channels therefore will also result in a depolarization, and like the results in spiny neurons, subthreshold depolarization resulted in an activation, i.e. increase in beating frequency. Full activation clamped the membrane potential at the reversal potential of chloride and thereby inhibited electrical activity in hiPSC-CMs. In this respect our findings with PSAM-GlyR and PSAM$^4$-GlyR activation do not come as a surprise and are also in line with results from recent work with optogenetic anion channels (Schwarzová et al., 2023).

## PSAM$^4$-GlyR superiority over PSAM-GlyR regarding agonist potency and agonist off-target effects

As described for neurons, the potency of varenicline for PSAM$^4$-GlyR was roughly 1000 times higher than of PSEM$^{89S}$ for PSAM-GlyR (Magnus et al., 2011, 2019). Micromolar PSEM$^{89S}$ concentrations were necessary to inhibit EHT contractility. While 100 μM stopped contractility instantaneously, 30 μM inhibited contractility

within 30–60 s. However, at this concentration PSEM$^{89S}$ prolonged AP duration also in control CMs. Ten micromolar PSEM$^{89S}$ prolonged APD$_{90}$ by 15 ms, and at 100 μM, APD$_{90}$ was prolonged by 50 ms. APD was prolonged at voltages below 0 mV, strongly suggesting $I_{Kr}$ block. To our knowledge, there are no data so far on the ability of PSEM$^{89S}$ to block hERG channels. From AP recordings in EHT, we estimate an effect on APD by PSEM$^{89S}$ similar in size as that between 3 and 10 nM of the HERG channel blocker E-4031 (Lemoine et al., 2018). Such an amount of hERG channel block should be associated with a substantial pro-arrhythmogenic risk, disqualifying any *in vivo* application of PSEM$^{89S}$. The situation is different for PSAM$^4$-GlyR. Varenicline-activated PSAM$^4$-GlyR at concentrations 50 times lower than those needed to block hERG channels (Armstrong et al., 2017).

In addition to the suspected hERG channel block by PSEM$^{89S}$, we expect another issue when using PSAM-GlyR hiPSC-CMs. PSAM$^4$-GlyR-expressing hiPSC-CMs had an inward current of ∼100 pA at −75 mV (varenicline absent). In contrast, currents were much higher in PSAM-GlyR (∼400 pA, again in the absence of an agonist). If we assume 1 pA/pF physiological chloride conductivity in CMs, a total chloride current of ∼40 pA per cell would result (Sakai et al., 1995). The remaining ∼60 pA in PSAM$^4$-GlyR can be

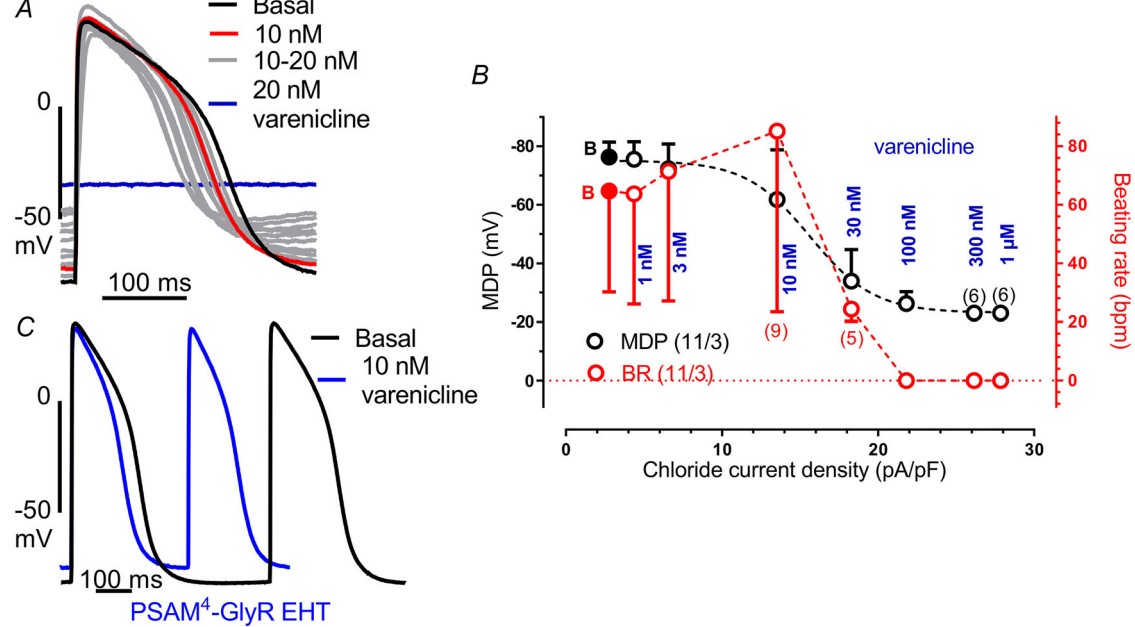

**Figure 6. Relationship between activated chloride current and AP parameters**

*A*, original action potential traces recorded in PSAM$^4$-GlyR EHTs in the presence of cumulatively increasing varenicline concentrations (1 nM steps up to 20 nM), illustrating the shift of the membrane potential. *B*, comparison between the chloride current density on the maximal diastolic potential (MDP) and beating rate in PSAM$^4$-GlyR EHT. Data are taken from patch clamp and sharp microelectrode measurements at matching varenicline concentrations. Note that 2 out of 11 EHTs stopped beating at 10 nM varenicline and 6 out of 11 EHTs stopped beating at 30 nM varenicline. Mean values ± SD for MDP (black) or beating rate (red). The number of EHTs/number of batches are given as *n/n*. *C*, maximum beating rate at 10 nM varenicline represented by two action potentials in PSAM$^4$-GlyR EHT. [Colour figure can be viewed at wileyonlinelibrary.com]

easily explained as leak current resulting from a realistic seal resistance of 1.25 GΩ (Heijman & Christ, 2023). Assuming the same seal resistance and physiological chloride conductivity as in PSAM$^4$-GlyR, we have to expect a constitutively active chloride current density of about ~300 pA that could explain the slightly less negative MDP in PSAM-GlyR. We cannot rule out that the less negative MDP in PSAM-expressing EHT may also result from some impairment of depolarizing potassium currents like $I_{K1}$ and $I_{Kr}$.

## Potential of PSAM$^4$-GlyR to modulate beating rate in CMs

We recently demonstrated that light activated anion-channels can be used to stop hiPSC-CMs with prolonged photo stimulation, but also to pace hiPSC-CMs with pulsed photo stimulation (Schwarzová et al., 2023). Light application in intact animals remains a technical issue. Activation of ion channels by drugs in contrast is a technically rather easy approach. At first glance, our results and the work Wexler et al. (2023) may be inter-

preted in a way that is similar to optogenetics: titrated activation of chemogenetics could be used to accelerate as well as inhibit CM activity. While we do see an application potential for inhibiting CM activity (also *in vivo*), we see substantial issues for the use of even PSAM$^4$-GlyR to accelerate CM activity. The work of Wexler et al. (2023) and this study indicate that titrated activation of PSAM$^4$ (either PSAM$^3$-5HT3 or PSAM$^4$-GlyR) can modulate beat rate in hiPSC-CMs. HiPSC-CMs in the EHT format show signs of maturation, i.e. physiological upstroke velocity and sodium current density, they still show signs of immaturity. Thus, (i) it remains open whether the same holds true for adult human CMs where cardiac inward rectifying potassium currents ($I_{K1}$) are much higher (Horváth et al., 2018) and are expected to oppose chloride current-induced depolarization. (ii) Even in case this strategy will work in adult CMs, the use of PSAM$^4$-GlyR and PSAM$^4$-5HT3 to accelerate beating rate carries substantial risks. Shifting the diastolic potential to a less negative potential brings inward channels closer to their activation threshold, facilitating depolarization (i.e. increasing activity). At the same time, steady-state inactivation of sodium channels takes place

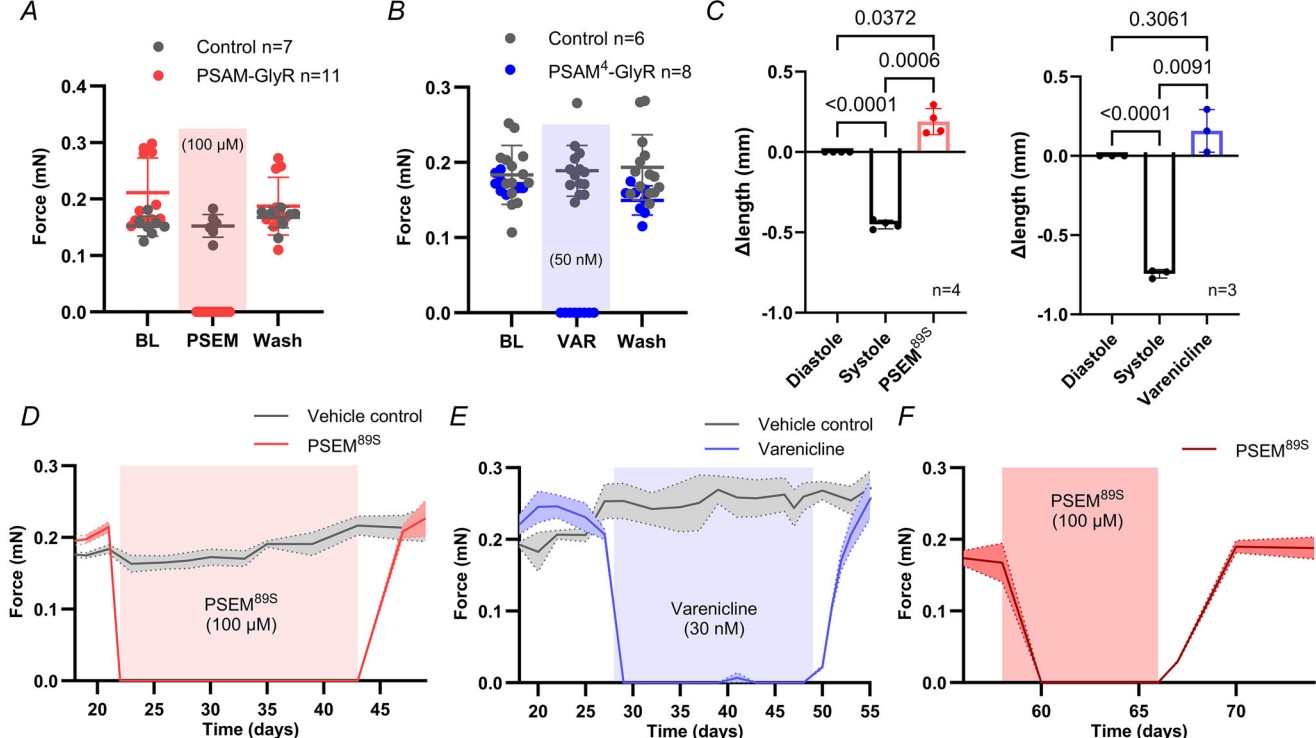

**Figure 7. PSAM-GlyR and PSAM$^4$-GlyR activation stopped contractility in engineered heart tissue**
*A*, video-optical contractility measurement of control (*n* = 7) and PSAM-GlyR EHT (*n* = 11). *B*, video-optical contractility measurement of control (*n* = 14) or PSAM$^4$-GlyR EHT (*n* = 8). *C*, change in EHT length from diastole to systole and during PSEM$^{89S}$ or varenicline treatment, respectively. Length was normalized to diastolic length. *n* = 4/3. One-way ANOVA with multiple comparisons. *P*-values are provided in the graph. *D–F*, contractility measurement during long-term silencing by constant PSEM$^{89S}$ (*D* and *F*) after a culture period of (*D*) ~1 month and (*F*) ~2 months or varenicline application (*E*; *n* = 4/3/3 EHTs per group). Mean values ± SD for contractile force (*A* and *B*) and EHT length (*C*). *n* indicates the number of EHTs. [Colour figure can be viewed at wileyonlinelibrary.com]

(Skibsbye et al., 2016) as illustrated by a drop in $V_{max}$ in our experiments. Since conduction velocity in cardiac tissue critically depends on the activity of sodium channels (Hondeghem & Katzung, 1980), the increase in beating rate will be associated with conduction slowing and thus may generate a substrate for arrhythmias. (iii) In addition, the range of beating rate modulation will be rather small. In hiPSC-CMs it ranged from 10 to 50 bpm for PSAM[4]-5HT3 (Wexler et al., 2023) or from 90 to 130 bpm in this study. (iv) Eventually, the range between a sub-threshold activation, i.e. an increase in beating frequency and an abrupt inhibition of beating is small. Thus, rather than for accelerating heart rate, PSAM[4]-GlyR should be a useful tool to stop CM activity.

## PTX did not block chloride currents through PSAM-GlyR and PSAM[4]-GlyR

A surprising finding of our study was the lack of an inhibitory effect of PTX on PSAM[4]-GlyR. Magnus et al. reported that PTX did not reduce basal currents (in the absence of varenicline) in PSAM[4]-GlyR-expressing cells, concluding that there was no constitutive activity of PSAM[4]-GlyR (Magnus et al., 2019). The evidence that PTX indeed inhibited chloride currents in PSAM[4]-GlyR comes from a single experiment, in which 1 mM PTX reduced an acetylcholine (300 nM)-stimulated chloride current by 70%. No data on the combination of the more specific agonist varenicline and PTX were reported (Magnus et al., 2019).

Our experiments do not support the assumption that PTX directly blocks PSAM-GlyR channels. PTX is used often in physiological research; however, the exact mode of action of PTX is not well understood and depends on the channel (sub)type investigated. It is often called a non-competitive ion pore antagonist. Biophysical experiments (estimation of the reversal potential in the presence and absence of extracellular chloride; Grutter et al., 2005) demonstrated that chloride ions are indeed the charge carrier of chimaeras between the ligand-binding domain of the nicotinergic acetyl-choline receptor and the ion pore domain of the GABA receptor. PTX antagonized GABA-induced chloride conductance in neurons. From the non-competitive mode of inhibition of GABA-induced chloride currents, the pore-forming domain was suspected to be a target for PTX (Newland & Cull-Candy, 1992). However, later sophisticated modelling excluded the possibility of an open-channel block as the mechanism of antagonism by PTX (Korshoej et al., 2010). This finding is in line with our study where we did not find evidence that PTX blocks chloride currents generated by PSAM[89S] and varenicline

with the chloride-selective ion pore domain of the glycine receptor $\alpha_1$ subunit being expressed.

Wexler et al. reported that PTX reversed the varenicline-induced depolarizing current in PSAM[4]-5HT3-expressing CMs. PSAM[4]-5HT3 constitutes a different form of PSAM, in which the pore-forming domain was derived from the 5-HT3 receptor, with $Na^+$ and $K^+$ as charge carriers. Partial reversal of the varenicline-induced depolarization in PSAM[4]-5HT3-expressing CMs in the study by Wexler et al. could be explained by an antagonism between varenicline and PTX at the pore-forming 5-HT3 receptor domain (Das & Dillon, 2005).

In this study we provide a comprehensive characterization of the chemogenetic tools PSAM-GlyR and PSAM[4]-GlyR in CM. We think that these tools are of interest because we can study the effects of chloride channel activation on action potential characteristics. Moreover, these tools might be of interest to inhibit contractility with high temporal resolution without (major) toxicity as indicated by the full recovery in force even after prolonged time periods. We cannot fully rule out a transient reactivation of electrical activity during prolonged off-switch periods but have no indication of this at present. Thus, they allow the study of aspects of cardiac biology that were difficult to address so far: (i) the effect of cardiac contractility on maturation, (ii) molecular consequences of unloading, and (iii) the interplay between contractility and metabolism. They could also serve as valuable tools in imaging studies to temporarily inhibit CM contractility. Additionally, even though far-fetched, PSAM[4]-GlyR could also find application in humans, e.g. in anti-arrhythmic gene therapy or to improve safety of pluripotent stem cell-derived CM transplantation and suppress engraftment-induced arrhythmias which represent a substantial hurdle towards the successful clinical translation of CM transplantation (Eschenhagen & Weinberger, 2024). Here one could speculate that a transient inhibition selectively of the engrafted calls could prevent engraftment-induced arrhythmias while not altering the electrophysiology of the host myocardium.

The tools described here could be successfully used to modulate chloride conductivity and contractility in stem-cell-derived CMs in the EHT format. While these CM are more mature than their 2D-counterparts they are still not fully matured. Whether PSAM-GlyR and PSAM[4]-GlyR can also be efficient in fully mature CMs (or stem cell-derived CMs that mature after trans-plantation) that possess different electrical properties and experience enhanced workload remains to be seen.

# Appendix

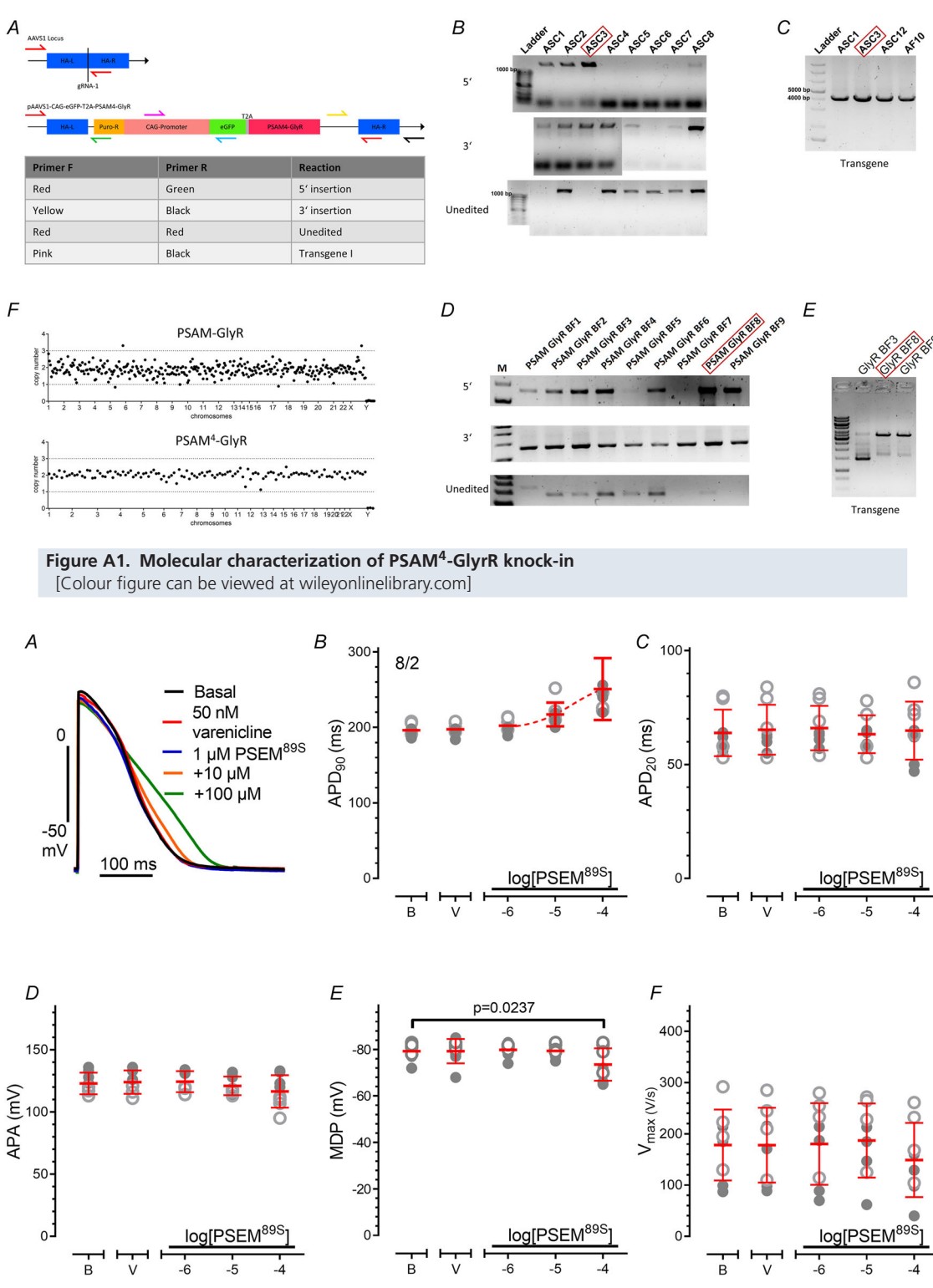

**Figure A1.  Molecular characterization of PSAM⁴-GlyrR knock-in**
[Colour figure can be viewed at wileyonlinelibrary.com]

**Figure A2.  PSEM⁸⁹ˢ but not varenicline prolonged action potential duration**
[Colour figure can be viewed at wileyonlinelibrary.com]

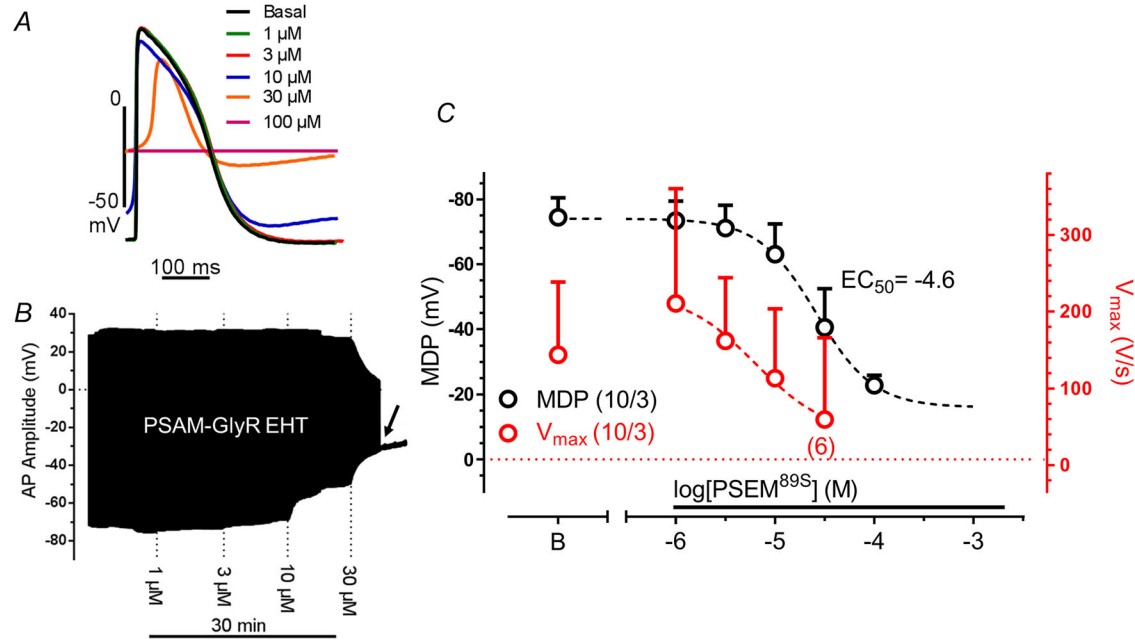

**Figure A3. Concentration-dependent effect of PSEM$^{89S}$ on AP characteristics in PSAM-GlyR EHT**
[Colour figure can be viewed at wileyonlinelibrary.com]

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

## Additional information

### Data availability statement

All data for this study is reported in the main manuscript or the Supplementary information respectively. The PSAM-GlyR and PSAM[4]-GlyR cell lines will be made available from the corresponding author via a material transfer agreement.

### Competing interests

The authors have no competing interests.

### Author contributions

The experiments for this study were performed at the Department of Experimental Pharmacology at the University Hospital Hamburg-Eppendorf. M.S. designed experiments, acquired data (electrophysiological experiments) and wrote the manuscript. T.S. designed experiments, acquired data (generation of the PSAM-GlyR and PSAM[4]-GlyR cell lines, contractility experiments), analysed data, acquired funding and wrote the manuscript. C.M. acquired data (histology and contractility experiments). A.C. acquired data (generation of the PSAM[4]-GlyR cell line). N.S. acquired data (stem cell culture, cardiomyocyte differentiation and generation of engineered heart tissue). J.I. acquired data (electrophysiological experiments). B.S., J.R., R.S. and A.S. acquired data (stem cell culture and cardiomyocyte differentiation and contractility experiments). E.S. acquired data (varenicline concentration measurements) and critically revised the content. T.E. critically revised the work. T.C. and F.W. conceived the study, designed experiments, acquired funding and wrote the manuscript. All authors have approved the final version of the manuscript and agree to be accountable for all aspects of the work in ensuring that questions related to the accuracy or integrity of any part of the work are appropriately investigated and resolved. All persons designated as authors qualify for authorship, and all those who qualify for authorship are listed.

## Funding

This work was supported by a research grant from the Werner Otto Stiftung (to T.S. and F.W.). This work has received funding from the European Union´s Horizon2020 FetOpen RIA (964800; to F.W.) and the European Union's Horizon 2020 research and innovation programme under the grant agreement No 874764 (Reanima to T.E.). M.S. was supported by the Philipp Schwartz Initiative of the Alexander von Humboldt Foundation and the Hamburg Programme for Scholars at Risk (HPSAR).

## Acknowledgements

The authors gratefully acknowledge the expert technical assistance of Anna Steenpaß. Authors thank all members of the induced pluripotent stem cell-derived cardiomyocyte working group at the Institute of Experimental Pharmacology and Toxicology, UKE-Hamburg, for their support with stem cell culture and hiPSC-CM differentiation.

## Keywords

cardiac electrophysiology, cardiac function, cardiomyocyte, chemogenetics, pluripotent stem cells

## Supporting information

Additional supporting information can be found online in the Supporting Information section at the end of the HTML view of the article. Supporting information files available:

**Peer Review History**

