## [Peer Review History · The Journal of Physiology]

Modulation of cardiomyocyte contractility and action potentials with chemogenetic chloride currents

Muhammed Sönmez, Tim Stüdemann, Christoph Manthey, Anita Covic, Nancy Shehata, Junsoo Im, Barbora Schwarzova, Judith Rössinger, Rajiven Srikantharajah, Aya Shibamiya, Edzard Schwedhelm, Thomas Eschenhagen, Torsten Christ, and Florian Weinberger

DOI: 10.1113/JP286428

Corresponding author(s): Florian Weinberger (floriananton.weinberger@cnic.es)

Review Timeline:

Submission Date:	14-Feb-2024
Editorial Decision:	14-Mar-2024
Revision Received:	17-Oct-2024
Editorial Decision:	11-Nov-2024
Revision Received:	26-Nov-2024
Editorial Decision:	13-Jan-2025
Revision Received:	17-Jan-2025
Accepted:	22-Jan-2025

Senior Editor: Kim Barrett

Reviewing Editor: Brian Delisle

Transaction Report:

Dear Dr Weinberger,

Re: JP-RP-2024-286428 "Modulation of cardiomyocyte contractility and action potentials with chemogenetic chloride currents" by Muhammed Sönmez, Tim Stüdemann, Christoph Manthey, Anita Covic, Nancy Shehata, Barbora Schwarzova, Judith Rössinger, Rajiven Srikantharajah, Aya Shibamiya, Edzard Schwedhelm, Thomas Eschenhagen, Torsten Christ, and Florian Weinberger

Thank you for submitting your manuscript to The Journal of Physiology. It has been assessed by a Reviewing Editor and by 2 expert referees and we are pleased to tell you that it is potentially acceptable for publication following satisfactory major revision.

REVISION CHECKLIST:

Please upload two versions of your manuscript text: one with all relevant changes highlighted and one clean version with no changes tracked. The manuscript file should include all tables and figure legends, but each figure/graph should be uploaded as separate, high-resolution files. The journal is now integrated with Wiley's Image Checking service. For further details,

see: <https://www.wiley.com/en-us/network/publishing/research-publishing/trending-stories/upholding-image-integrity-wileys-image-screening-service>

We look forward to receiving your revised submission.

Yours sincerely,

Kim Barrett
Senior Editor
The Journal of Physiology

REQUIRED ITEMS FOR REVISION

- Author photo and profile. First or joint first authors are asked to provide a short biography (no more than 100 words for one author or 150 words in total for joint first authors) and a portrait photograph. These should be uploaded and clearly labelled together in a Word document with the revised version of the manuscript. See Information for Authors for further details.

- The reference list must be in alphabetical order, rather than numbered, to comply with our Journal format.

- Your manuscript must include a complete Additional Information section, including competing interests; funding; author contributions and acknowledgements.

- Please upload separate high-quality figure files via the submission form.

- Papers must comply with the Statistics Policy: https://jp.msubmit.net/cgi-bin/main.plex?form_type=display_requirements#statistics.

In summary:

- If $n \leq 30$, all data points must be plotted in the figure in a way that reveals their range and distribution. A bar graph with data points overlaid, a box and whisker plot or a violin plot (preferably with data points included) are acceptable formats.

- If $n > 30$, then the entire raw dataset must be made available either as supporting information, or hosted on a not-for-profit repository, e.g. FigShare, with access details provided in the manuscript.

- 'n' clearly defined (e.g. x cells from y slices in z animals) in the Methods. Authors should be mindful of pseudoreplication.

- All relevant 'n' values must be clearly stated in the main text, figures and tables.

- The most appropriate summary statistic (e.g. mean or median and standard deviation) must be used. Standard Error of the Mean (SEM) alone is not permitted.

- Exact p values must be stated. Authors must not use 'greater than' or 'less than'. Exact p values must be stated to three significant figures even when 'no statistical significance' is claimed.

- A Data Availability Statement is required for all papers reporting original data. This must be in the Additional Information section of the manuscript itself. It must have the paragraph heading 'Data Availability Statement'. All data supporting the results in the paper must be either: in the paper itself; uploaded as Supporting Information for Online Publication; or archived in an appropriate public repository. The statement needs to describe the availability or the absence of shared data. Authors must include in their statement: a link to the repository they have used, or a statement that it is available as Supporting Information; reference the data in the appropriate section(s) of their manuscript; and cite the data they have shared in the References section. Whenever possible, the scripts and other artefacts used to generate the analyses presented in the paper should also be publicly archived. If sharing data compromises ethical standards or legal requirements then authors are not expected to share it, but must note this in their statement. For more information, see our Statistics Policy.

- Please include an Abstract Figure file, as well as the Figure Legend text within the main article file. The Abstract Figure is a piece of artwork designed to give readers an immediate understanding of the research and should summarise the main conclusions. If possible, the image should be easily 'readable' from left to right or top to bottom. It should show the physiological relevance of the manuscript so readers can assess the importance and content of its findings. Abstract Figures should not merely recapitulate other figures in the manuscript. Please try to keep the diagram as simple as possible and without superfluous information that may distract from the main conclusion(s). Abstract Figures must be provided by authors no later than the revised manuscript stage and should be uploaded as a separate file during online submission labelled as File Type 'Abstract Figure'. Please also ensure that you include the figure legend in the main article file. All Abstract Figures should be created using BioRender. Authors should use The Journal's premium BioRender account to export high-resolution images. Details on how to use and access the premium account are included as part of this email.

EDITOR COMMENTS

Reviewing Editor: Comments for Authors to ensure the paper complies with the Statistics Policy:
Please review and modify the manuscript to comply with the Journal's statistic's policy https://jp.msubmit.net/cgi-bin/main.plex?form_type=display_requirements#statistics

Comments to the Author:

Both referees agree that the manuscript represents a technological advancement, but they are split on whether they think the work will have a major impact on the field. Referee 1 raised several concerns that will require more experiments and clarity in the interpretation and presentation of the results. Although Referee 2 was more enthusiastic, they expressed similar concerns about the presentation. A manuscript that satisfactorily addresses all of the referees' concerns, including concerns about sample size, interpretation/presentation of the results, and clarifying any potential clinical relevance these findings have, may elevate the work to have a significant impact.

Senior Editor:

Comments for Authors to ensure the paper complies with the Statistics Policy:
Please review our statistics policy. Note specifically that SEM is not permitted as a measure of variability.

REFEREE COMMENTS

Referee #1:

Summary

The authors provide data in hiPSC-CMs expressing PSAM peptides as a mechanism to modulate cardiac electrical activity. The authors provide substantial data that suggest PSAM activation would be an effective tool to modulate (more specifically halt) cardiac electrical activity. PSAM activation appears to depolarize the resting membrane potential, ultimately triggering depolarization block and the cessation of action potential firing. That said there are several significant limitations within the manuscript that should be addressed. Some sample sizes are far too small to provide the necessary statistical power in a hiPSC-CM model and a key control is lacking in the Figure 5 experiment.

Specific comments

Line 84: "Patch clamp recordings demonstrated that channel activation resulted in chloride-driven inward currents that depolarized the cell." This is incorrect, an inward Cl⁻ current would hyperpolarize the cell. Cl⁻ efflux depolarizes the cell.

What was the voltage error?

Why does PSAM4-GlyR display bimodal distribution in Figure 1?

Line 154: "PSAM-GlyR and PSAM4-GlyR EHT developed regularly" this is abroad statement that requires quantitative data to back it up. Also, given the reported differences in membrane potential, this may not be true.

Sample sizes for PSAM4-GlyR in Figure 2 are inadequate for reliable statistical power given the known variability in hiPSC-CMs.

Sample sizes in Figure 3 are too small to draw conclusions and should be increased. The data do suggest, however, that the PSAMs are effecting other currents, such as IKr or IK1. This should be determined by recording E-4031 and barium-sensitive currents, respectively.

Figure 5 lacks appropriate vehicle controls.

The applicability of the model to increase beating frequency seems misconstrued. This would only be relevant in nodal tissue. Applying this technique to atrial or ventricular tissue would merely cause arrhythmia. Additionally, the rationale that one would want to stop cardiac activity is unclear.

Referee #2:

Many thanks for inviting my review of this piece of work from Sönmez et al which documents the application of a chemogenetic strategy to manipulate the iPS-CM in the engineered heart context.

The chemigenetic toolbox is relatively under utilised in cardiac biology, but the clarity with which the authors have evaluated their chosen tools in this submission should serve the as a useful reference for anyone wishing to follow suit.

Once again this group appear to identify what others (<https://elifesciences.org/digests/38506/the-chloride-paradox>) have termed "the chloride paradox" - where something that should make depolarisation less likely, for some reason shows the opposite biological outcome. There are now several such examples.

Overall, I felt the paper was well written, and an appropriate length.

I have some comments which may improve the article further if they could be addressed.

MAJOR COMMENTS:

1) A CRISPr knock-in strategy was adopted. 85% of differentiated cells expressed the PSAM, therefore up to 15% did not. Could the authors enlighten on:

a. The average number of integrated PSAM's per genome (and whether in clone selection a threshold number was seen for

the biological effects reported)

b. Whether the absence of PSAM expression represents gene expression at the margins of detection, random silencing (PMID: 32789000), or clonal heterogeneity? Does this underpin the apparent differences in drug response (eg 3/5 EHT stop with 30nM varenicline, Figure 5).

c. Could the authors explain how all cells in the EHT will follow the PSAM activation even if not all cells express the transgene; and perhaps more importantly the extension to that: why in an engraftment context stopping the graft would not lead to stopping the heart?

My request for this is motivated by the difficulties in reproducibility that will follow if equivalent transgene expression levels cannot be reached by others.

2) Would the authors be able to speculate on whether the iPS-CM, with its distinct electrophysiological properties, is more amenable to PSAM based modulation compared to an adult ventricular cardiomyocyte for example?

My motivation for this question is trying to understand what may happen in the event the strategy were applied to engraftment arrhythmia (which I agree would definitely be worth testing based on this data).

As the graft matures in vivo (as a minimum cell size increases (PMID: 28065644, PMID: 35089805) alongside electrical properties PMID: 31056479, PMID: 29969440) will the PSAM continue to behave as described in this work? What would be the consequence of heterogeneity in the maturation process within a graft?

Of course, acknowledging the authors prior work demonstrating the role pacing may play in EHT maturation (PMID: 24852842), I fully accept that structural and electrical maturation may occur differently if the graft is held depolarised by PSAM activation (Fig 7D), but is there an inherent risk that this strategy will defer a problem rather than fix it?

Would it be possible to use a computational modelling approach (eg PMID: 31868580) to explore this (now or in future work)?

3) The most striking observation in the manuscript is the ability to hold the iPS-CM EHT in a non-force producing state for nearly a month (Figure 7D). I think the reader would expect to be shown whether the resting length of the EHT was maintained in the systolic equivalent of this model (as predicted by the depolarisation reported), whether there was any evidence of cyclical electrical activity, and whether there was any impact on sarcomere integrity or gap junction formation which may underpin the almost week long recovery process in force production following varenicline wash-out. Presence/absence of apoptotic induction would also appear relevant.

4) I hope I have not missed a supplementary methods section, but as it stands the methods section is surprisingly light on detail.

The starting iPS line is not identified in the current methods, passage number and baseline genome integrity are not commented on. Following gene knock-in it is not clear what clone characterisation has been undertaken to address genome integrity in particular.

The analysis of contractile force by video-optical recording is lacking either a reference, or some detail.

Bizarrely, even the source and the preparation of the small molecules (particularly picrotoxin - which did not work as expected, varenicline, and PSEM89S) evaluated in the chemigenetic approach are not described. This is not acceptable.

Minor comments:

In Figure 2 reference is made to 8 EHT's in the PSAM group, however there are only seven red circles in each of the figures.

There appears to be a discrepancy between the beat frequency presented in fig 1E (50-70bpm, with the statistically different and higher number attributed to PSAM, and the lower to the WT EHTs) and the cycle length data presented in figure 2D - 700-900msec. Please comment.

In the manuscript the language about hiPS-CM or EHT is a little loose, it would probably help the reader if the nomenclature could use EHT for EHT assays, and iPS-CM for single cell aspects.

Please clarify the meaning of "sensitive" in the following sentence "In fact, in neurons PSAM4-GlyR was much more sensitive than PSAM-GlyR" (Line 230)

I wonder if it would help the general reader understand the role of chloride currents in the heart in general (eg PMID: 9876314) and why some of the previously articulated concerns about endogenous chloride channel activation may (or may not) apply to the exogenous transgene used in this study.

END OF COMMENTS

EDITOR COMMENTS

Reviewing Editor: Comments for Authors to ensure the paper complies with the Statistics Policy:

Please review and modify the manuscript to comply with the Journal's statistic's policy https://jp.msubmit.net/cgi-bin/main.plex?form_type=display_requirements#statistics

Comments to the Author:

Both referees agree that the manuscript represents a technological advancement, but they are split on whether they think the work will have a major impact on the field. Referee 1 raised several concerns that will require more experiments and clarity in the interpretation and presentation of the results. Although Referee 2 was more enthusiastic, they expressed similar concerns about the presentation. A manuscript that satisfactorily addresses all of the referees' concerns, including concerns about sample size, interpretation/presentation of the results, and clarifying any potential clinical relevance these findings have, may elevate the work to have a significant impact.

Following the reviewers' suggestions we have performed additional experiments to increase the number of biological replicates for the contractility measurements, the action potential recordings and the patch clamp analysis (new Figures 1-3). These experiments confirmed our previous results and substantiated our findings. We have also included more information on the generation of the PSAM4-GlyR cell line (new Supplementary Figure1) as well as information on the tissue length during depolarization block (new Figure 7C). Additionally, we have modified the discussion to clarify our results and better describe the impact for the field.

Senior Editor:

Comments for Authors to ensure the paper complies with the Statistics Policy:

Please review our statistics policy. Note specifically that SEM is not permitted as a measure of variability.

We have adjusted this and now provide SD.

Referee #1:

Summary

The authors provide data in hiPSC-CMs expressing PSAM peptides as a mechanism to modulate cardiac electrical activity. The authors provide substantial data that suggest PSAM activation would be an effective tool to modulate (more specifically halt) cardiac electrical activity. PSAM activation appears to depolarize the resting membrane potential, ultimately triggering depolarization block and the cessation of action potential firing. That said there are several significant limitations within the manuscript that should be addressed. Some sample sizes are far too small to provide the necessary statistical power in a hiPSC-CM model and a key control is lacking in the Figure 5 experiment.

We would like to thank the reviewer for his/her work on our manuscript. We agree with the reviewer regarding sample size and have performed new experiments, increased the number of biological replicates and substantiated our findings.

Specific comments

Line 84: "Patch clamp recordings demonstrated that channel activation resulted in chloride-driven inward currents that depolarized the cell." This is incorrect, an inward Cl⁻ current would hyperpolarize the cell. Cl⁻ efflux depolarizes the cell.

We thank the reviewer for giving us the opportunity to discuss effect of Cl⁻ fluxes on membrane potential. The actual membrane potential accounts as " $E_m = g'_K (E_K) + g'_{Na} (E_{Na}) + g'_{Ca} (E_{Ca}) + g'_{Cl} (E_{Cl})$ " (Goldman 1943). Therefore, any increase in the conductivity for a given ion will bring the membrane potential closer to its equilibrium potential. Exact reversal potential (E_{rev}) for Cl⁻ under our conditions is hard to judge, since [Cl]_i in hiPSC-CM is not known. However, there is no doubt that the reversal potential for Cl⁻ is positive from E_{rev} for K⁺ and negative from E_{rev} for Na⁺. Thus, any increase in Cl⁻-conductance in diastole (dominated by E_{rev} for K⁺ and negative from E_{rev} for Na⁺) would bring the diastolic potential (-80 mV in our case) to more positive values (depolarization). The same intervention (increase in Cl⁻-conductance) during plateau phase would bring membrane potential to more negative values (hyperpolarization). Thus, compared with cationic channels, cardiac Cl⁻ channels have the unique ability to generate both inward and outward currents and cause both depolarization and repolarization during the action potential (Duan 2013).

Please also see the comment of referee #2, who pointed exactly to this apparent paradox.

What was the voltage error?

Thanks for this comment. We have calculated the voltage error by using the following equations (Gray and Santin 2023):

- Effective resistance (R_{eff} , M Ω) = Seal resistance (R_s) x (1- %Comp/100)
- Voltage error = $V_{Pipette} - V_{Membrane} = I_f$ (feedback current) x R_{eff}

Calculated values for every current size are given for different substance concentration in the following table(below). Even with the highest Cl⁻ currents activated the maximum voltage error was not larger than 1 mV. This information is now given in the section methods (lines 138-141):

"We have calculated voltage errors in our setting by the following equation:

Effective resistance (R_{eff} , M Ω)= Seal resistance (R_s) x (1- %Comp/100)

Voltage error= $V_{Pipette} - V_{Membrane} = I_f$ (feedback current) x R_{eff}

We measured a R_s of 4.9 ± 0.1 (M Ω) n= 134 that was compensated by 80 % giving a R_{eff} of 0.98 (M Ω). Even with larger Cl⁻ current amplitudes resulting voltage error was not larger than 1 mV."

	I_F (nA)	Voltage Error (mV)
PSAM-GlyR (6/3)		
Log [PSEM⁸⁹⁵] (M)		
Basal	0.3±0.1	0.3±0.2
-6.00	0.4±0.3	0.4±0.3
-5.50	0.4±0.3	0.4±0.3
-5.00	0.6±0.3	0.5±0.3
-4.50	0.7±0.2	0.7±0.2
-4.00	1.0±0.4	1.0±0.4
PSAM⁴-GlyR (13/4)		
Log [varenicline (M)]		
Basal	0.1±0.0	0.1±0.0
-9.00	0.2±0.1	0.2±0.1
-8.50	0.3±0.1	0.3±0.1
-8.00	0.5±0.2	0.5±0.2
-7.50	0.7±0.2	0.6±0.2
-7.00	0.8±0.2	0.8±0.2
*R _s = 4.9±0.1 (MΩ) n= 134		
*R _{eff} = 0.98 (MΩ) % 80 compensation		

Why does PSAM4-GlyR display bimodal distribution in Figure 1?

There was indeed batch-to batch variation which is something that we encounter regularly with stem cell-derived CM and which is often greater with gene edited CM. We have now included another batch of EHTs and have i) rephrased the respective paragraph to better describe that genetically edited EHT showed greater variability (new lines 169-170) and ii) use different symbols for the respective batches to allow for a discrimination of individual batches (new Figure 1).

Line 154: "PSAM-GlyR and PSAM4-GlyR EHT developed regularly" this is abroad statement that requires quantitative data to back it up. Also, given the reported differences in membrane potential, this may not be true.

We have rephrased this paragraph to better more precisely describe what we intended to say, i.e. that EHT remodeled and coherently started to beat.

Sample sizes for PSAM4-GlyR in Figure 2 are inadequate for reliable statistical power given the known variability in hiPSC-CMs.

We performed new experiments and increased the number of experiments.

Sample sizes in Figure 3 are too small to draw conclusions and should be increased. The data do suggest, however, that the PSAMs are effecting other currents, such as I_{Kr} or I_{K1} . This should be determined by recording E-4031 and barium-sensitive currents, respectively.

Thanks for the comment regarding too small n numbers. We increased our data from 5/2/1 to 17/3/2 in PSAM⁴-hiPSC-CM (cell/EHT/batch, respectively). We agree with the reviewer that the less negative MDP in PSAM does not necessarily result from some leak Cl^- -conductivity. In principle, effects of PSAM expression on I_{K1} and I_{Kr} could account. To test this, we expect a large effort necessary. Human iPSC-CM are much smaller than adult CM making measurements of small potassium outward currents like I_{K1} and I_{Kr} technically demanding (Horváth et al. 2018). On the other side, the relevance of PSAM expression on I_{K1} and I_{Kr} would be minor at best, since we do not see a way how such interaction could be avoided or compensated. In addition, off target effects PSEM⁸⁹⁵ speaks against application of PSAM even more since PSAM⁴ represents an attractive alternative. We therefore added the following sentence at the end of the section "*PSAM4-GlyR superiority over PSAM-GlyR regarding agonist sensitivity and agonist off-target effects*" (lines 314-315):

"We cannot rule out that the less negative MDP in PSAM-expressing EHT may also result from some impairment of depolarizing potassium currents like I_{K1} and I_{Kr} ."

Figure 5 lacks appropriate vehicle controls.

We have measured AP continuously. From the recordings, we could identify a clear association between drug exposures and effect. Please note that pre-drug recordings and recordings at low, ineffective concentration of varenicline over 30 minutes remained stable in Figure 5.

The applicability of the model to increase beating frequency seems misconstrued. This would only be relevant in nodal tissue. Applying this technique to atrial or ventricular tissue would merely cause arrhythmia. Additionally, the rationale that one would want to stop cardiac activity is unclear.

We respectfully disagree. Wexler et al. 2023 commented with regard to application possibilities: "Given the bidirectional effects achieved, increasing automaticity in a dose-dependent manner at low Varenicline concentrations and suppressing automaticity and excitability at higher doses, this chemogenetic strategy could be utilized in a wide spectrum of applications. These include fine-tuned biological pacemaking approaches by genetically-modifying native cardiac cells or by transplanting engineered cells". As this is the only manuscript describing similar chemogenetic tools in cardiomyocytes, we think that it is of importance to discuss this aspect. In our opinion, the silencing aspect is attractive, because such genetically modified cells could be used directly for cell replacement therapies as currently performed by us and others. In contrast, we currently do not see a way to safely apply PSAM-GlyR for increasing beating frequency (even when targeting specific nodal tissue or transplanting genetically modified cells as pacemakers).

We agree that the application of chemogenetic tools might theoretically trigger arrhythmias, but this would need to be addressed experimentally. We (and several collaboration partners) are currently using this tool to stop cardiac contractility to i) assess the effect of cardiac work on cell cycle activity, ii) study the impact of cardiac contractility on cardiac metastases, iii) apply it in imaging studies to avoid motion artefacts and iv) assess its value in the suppression of engraftment-induced arrhythmias. As these studies are all preliminary and ongoing, we cannot discuss them in detail but in accordance to Wexler et al. and Reviewer 2 we believe that chemogenetic tools are valuable in cardiovascular biology.

Referee #2:

Many thanks for inviting my review of this piece of work from Sönmez et al which documents the application of a chemogenetic strategy to manipulate the iPS-CM in the engineered heart context.

The chemigenetic toolbox is relatively under utilised in cardiac biology, but the clarity with which the authors have evaluated their chosen tools in this submission should serve the as a useful reference for anyone wishing to follow suit.

We would like to thank the reviewer for his/her work on our manuscript and the kind words regarding the quality of our work. Please find a point-to point reply below.

Once again this group appear to identify what others (<https://elifesciences.org/digests/38506/the-chloride-paradox>) have termed "the chloride paradox" - where something that should make

depolarisation less likely, for some reason shows the opposite biological outcome. There are now several such examples.

Overall, I felt the paper was well written, and an appropriate length.

I have some comments which may improve the article further if they could be addressed.

MAJOR COMMENTS:

We would like to thank the reviewer for these comments. Indeed, the reviewer mentions several aspects that we are currently working on. These studies comprise in vitro work, small animal but also large animal work and, in our opinion, go far beyond the scope of this manuscript. However, we have expanded the discussion with regard to some of these aspects.

1) A CRISPR knock-in strategy was adopted. 85% of differentiated cells expressed the PSAM, therefore up to 15% did not. Could the authors enlighten on:

a. The average number of integrated PSAM's per genome (and whether in clone selection a threshold number was seen for the biological effects reported)

Our CRISPR/Cas9 strategy relies on homologous recombination of the plasmid with the genomic DNA locus. The roughly 800 base pair long homology arms, located upstream and downstream of the transgene are required for this process and make integration of multiple copies highly unlikely. We have genotyped the clones with Sanger sequencing (PSAM-GlyR and PSAM4-GlyR) and Southern Blot (PSAM-GlyR). These analyses demonstrated integration of a single copy in the clones used for this study. We did not follow up on clones that did not reveal a correct integration of a single copy. While the generation of the PSAM-GlyR cell line is already described in detail in Stüdemann et al. *Circulation* 2022, we have now added information on the generation of the PSAM⁴-GlyR cell line (new Supplementary Figure S1). While we have not systematically studied effects of PSEM⁸⁹⁵ and varenicline with different clones we have seen the main effect, inhibition of contractility in cardiomyocytes derived from two different clones per cell line (each clone carrying a single gene copy per allele).

b. Whether the absence of PSAM expression represents gene expression at the margins of detection, random silencing (PMID: 32789000), or clonal heterogeneity? Does this underpin the apparent differences in drug response (eg 3/5 EHT stop with 30nM varenicline, Figure 5).

We have stated that transgene expression was greater than 85% based on the iPSC data. The FACS data for cardiomyocytes in contrast showed that 90% of all cells expressed GFP. We now went back to our original data and systematically analyzed the FACS data for iPSC and cardiomyocytes over time. We saw that ~85% represented the lowest example but most iPSC samples showed GFP expression around 90%. GFP expression in cardiomyocytes was 93% on average. Neither for PSAM-GlyR nor for PSAM⁴-GlyR could we detect a decrease in GFP expression over time. We have also repeated the genotyping and saw no indication for clonal heterogeneity. In summary, we think that the incomplete GFP expression does not

result from clonal heterogeneity but rather random silencing. We are well aware of the manuscript that the reviewer mentions (and indeed contacted the senior author when we first started our knock-in approaches). However, in our hands silencing has not been a major problem so far. We have by now generated about ten cell lines in which we knocked-in opto- and chemogenetic tools in the AAVS1 locus. While there were slight differences in transgene expression, we have seen high transgene expression (more than 85% in all lines). While there were differences in the sensitivity for PSEM⁸⁹⁵ and varenicline the differences were small, and EHTs that stopped at lower varenicline concentrations were generated from the same differentiation run as EHT that did not stop with 30 nM varenicline, indicating that the different drug responses did not result from a heterogeneous transgene expression. We now discuss this aspect in lines 268-273.

c. Could the authors explain how all cells in the EHT will follow the PSAM activation even if not all cells express the transgene; and perhaps more importantly the extension to that: why in an engraftment context stopping the graft would not lead to stopping the heart?

As described above, transgene expression was (very) high in the hiPSC-CM (>90%). Thus, most likely we inhibited the CM responsible for pacemaking. Additionally, the very high transgene expression also inhibited impulse propagation. Importantly, EHTs act as an electrical syncytium and inhibition of a vast majority of the cells therefore inhibits ETH contractility in total. We did not see any ongoing micro-contraction in this setting.

With regard to the transplantation context, a chemogenetic strategy would selectively (temporarily) inhibit the transplanted cardiomyocytes expressing PSAMs but not the unmodified host cardiomyocytes and thus will not stop the heart. We consider this the important advantage of such strategy.

My request for this is motivated by the difficulties in reproducibility that will follow if equivalent transgene expression levels cannot be reached by others.

We can only speculate about the results from an approach with a virus-mediated overexpression targeting only a subpopulation of the cells but believe our approach has a great advantage in this regard. We have already shared the cell lines with other scientists and we are happy to share them with the scientific community in the future.

2) Would the authors be able to speculate on whether the iPS-CM, with its distinct electrophysiological properties, is more amenable to PSAM based modulation compared to an adult ventricular cardiomyocyte for example?

We can only speculate about this and discuss our ideas in lines 326-338.

My motivation for this question is trying to understand what may happen in the event the strategy were applied to engraftment arrhythmia (which I agree would definitely be worth testing based on this data).

We are currently working on this aspect. However, so far all data indicates that the engraftment-induced arrhythmias result from automatism of the transplanted cardiomyocytes. Therefore one would have to silence "only" the hiPSC-derived cardiomyocytes.

As the graft matures in vivo (as a minimum cell size increases (PMID: 28065644, PMID: 35089805) alongside electrical properties PMID: 31056479, PMID: 29969440) will the PSAM continue to behave as described in this work? What would be the consequence of heterogeneity in the maturation process within a graft?

Unfortunately, we can only speculate as we have not analyzed it in detail yet. We have transplanted PSAM-GlyR (Stüdemann et al. *Circulation* 2022) and PSAM⁴-GlyR cardiomyocytes (unpublished). We did not see any GFP negative CMs in the grafts four to eight weeks after transplantation. We also saw no apparent differences in graft maturation within these grafts. Therefore even when considering a small subpopulation that did not express the transgene, we think that it had no effect on graft maturation. Regarding the application for suppressing engraftment-induced arrhythmias, we also can only speculate but believe that there is a good rationale why it will not be a major problem. These arrhythmias are the consequence of a very large number of cardiomyocytes. When smaller numbers of cardiomyocytes were transplanted, no arrhythmias occurred (i.e. Kobayashi *Circulation* 2024), indicating that even when a small subpopulation will not be silenced it will not cause arrhythmias. However, at present this remains a speculation. While we think that the reviewer raises an important question, this goes far beyond the current manuscript in which we physiologically characterize the chemogenetic tools.

Of course, acknowledging the authors prior work demonstrating the role pacing may play in EHT maturation (PMID: 24852842), I fully accept that structural and electrical maturation may occur differently if the graft is held depolarised by PSAM activation (Fig 7D), but is there an inherent risk that this strategy will defer a problem rather than fix it?

We can fully understand this concern. Indeed, this was our initial hypothesis as well. We have preliminary, but convincing evidence that EHT mature in the absence of beating (unpublished, Shehata et al. Gordon Research Conference 2024). This is ongoing work and it is too early to report this data. Ongoing studies will show whether we solve the problem of engraftment-induced arrhythmias or simply defer it. We think that it goes far beyond the current manuscript.

Would it be possible to use a computational modelling approach (eg PMID: 31868580) to explore this (now or in future work)?

We agree with reviewer that computer modelling could be a useful tool to better understand action of artificial Cl⁻ channels. We have started to work on this and will explore this opportunity in future work.

3) The most striking observation in the manuscript is the ability to hold the iPS-CM EHT in a non-force producing state for nearly a month (Figure 7D). I think the reader would expect to be shown whether the resting length of the EHT was maintained in the systolic equivalent of this model (as predicted by the depolarisation reported), whether there was any evidence of cyclical electrical activity, and whether there

was any impact on sarcomere integrity or gap junction formation which may underpin the almost week long recovery process in force production following varenicline wash-out. Presence/absence of apoptotic induction would also appear relevant.

We now provide this information in line 253 and the new Figure 7E. In contrast to the assumption by the reviewer a depolarization block, such as seen with high-potassium cardioplegic solutions arrests cardiomyocytes in diastole (Melrose et al. The Lancet 1955). We have no means to assess electrical activity continuously over time. However, we have monitored EHTs daily during off-switch and did not detect any contractions, i.e. also on an individual cell level in microscopy. We think that the full recovery argues against a major toxicity and discuss this in lines 367 and 368. We agree that the effect on sarcomere integrity and gap junction formation is of interest and assess this in ongoing studies but again think that this is beyond the focus of the current manuscript.

4) I hope I have not missed a supplementary methods section, but as it stands the methods section is surprisingly light on detail.

The starting iPS line is not identified in the current methods, passage number and baseline genome integrity are not commented on. Following gene knock-in it is not clear what clone characterisation has been undertaken to address genome integrity in particular.

We have expanded this section and included the information (lines 85-107).

The analysis of contractile force by video-optical recording is lacking either a reference, or some detail.

We have added a reference.

Bizarrely, even the source and the preparation of the small molecules (particularly picrotoxin - which did not work as expected, varenicline, and PSEM89S) evaluated in the chemigenetic approach are not described. This is not acceptable.

We have included this information.

Minor comments:

In Figure 2 reference is made to 8 EHT's in the PSAM group, however there are only seven red circles in each of the figures.

We would like to thank the reviewer for pointing out this mistake. The Y-axis was cut off and therefore one data point was missing. We corrected our graph with the increased n-numbers.

There appears to be a discrepancy between the beat frequency presented in fig 1E (50-70bpm, with the statistically different and higher number attributed to PSAM, and the lower to the WT EHTs) and the cycle length data presented in figure 2D - 700-900msec. Please comment.

The reviewer is correct. The difference may relate to methodological differences when beating rat was measured. Figure 1 shows beating rate measured in culture medium, whereas Figure 2 reports data obtained Tyrode's solution. In order to avoid confusion of the potential reader we have added commented on this in the section results.

In the manuscript the language about hiPS-CM or EHT is a little loose, it would probably help the reader if the nomenclature could use EHT for EHT assays, and iPS-CM for single cell aspects.

We have adjusted the language in this respect.

Please clarify the meaning of "sensitive" in the following sentence "In fact, in neurons PSAM4-GlyR was much more sensitive than PSAM-GlyR" (Line 230).

We have rephrased this sentence.

I wonder if it would help the general reader understand the role of chloride currents in the heart in general (eg PMID: 9876314) and why some of the previously articulated concerns about endogenous chloride channel activation may (or may not) apply to the exogenous transgene used in this study.

We have included some information on chloride channels in the introduction (Line 58).

Dear Dr Weinberger,

Re: JP-RP-2024-286428R1 "Modulation of cardiomyocyte contractility and action potentials with chemogenetic chloride currents" by Muhammed Sönmez, Tim Stüdemann, Christoph Manthey, Anita Covic, Nancy Shehata, Junsoo Im, Barbora Schwarzova, Judith Rössinger, Rajiven Srikantharajah, Aya Shibamiya, Edzard Schwedhelm, Thomas Eschenhagen, Torsten Christ, and Florian Weinberger

Thank you for submitting your revised Research Article to The Journal of Physiology. It has been assessed by the original Reviewing Editor and Referees and has been well received. Some final revisions have been requested.

REVISION CHECKLIST:

We look forward to receiving your revised submission.

Yours sincerely,

Kim Barrett
Senior Editor
The Journal of Physiology

REQUIRED ITEMS

- Papers must comply with the Statistics Policy: https://jp.msubmit.net/cgi-bin/main.plex?form_type=display_requirements#statistics.

In summary:

- If $n \leq 30$, all data points must be plotted in the figure in a way that reveals their range and distribution. A bar graph with data points overlaid, a box and whisker plot or a violin plot (preferably with data points included) are acceptable formats.
- If $n > 30$, then the entire raw dataset must be made available either as supporting information, or hosted on a not-for-profit repository, e.g. FigShare, with access details provided in the manuscript.
- 'n' clearly defined (e.g. x cells from y slices in z animals) in the Methods. Authors should be mindful of pseudoreplication.
- All relevant 'n' values must be clearly stated in the main text, figures and tables.
- The most appropriate summary statistic (e.g. mean or median and standard deviation) must be used. Standard Error of the Mean (SEM) alone is not permitted.
- Exact p values must be stated. Authors must not use 'greater than' or 'less than'. Exact p values must be stated to three significant figures even when 'no statistical significance' is claimed.

NB- !!Please include the Abstract Figure Legend text within the main article file.

EDITOR COMMENTS

Reviewing Editor:

Comments for Authors to ensure the paper complies with the Statistics Policy:
Please review the journal's statistical policy on reporting the data https://jp.msubmit.net/cgi-bin/main.plex?form_type=display_requirements#statistics This include use of SEM and reporting of P-values.

Comments to the Author:

The manuscript has been re-reviewed by the original referees. Both referees conclude the manuscript will have a moderate to significant impact on the field. However, there are several questions and points that each referee raised that should be addressed. The authors are encouraged to address all these points and ensure the manuscript complies with the Journal's statistical policy.

Senior Editor:

Comments for Authors to ensure the paper complies with the Statistics Policy:
See comments from RE

REFEREE COMMENTS

Referee #1:

The authors have made a number of useful additions to the manuscript that improve it greatly the quality of the manuscript. There remain only two issues that should be addressed.

Your sentence, "...resulted in chloride-driven inward currents that depolarized the cell." is incorrect. Although I thoroughly enjoyed the in-depth review of my undergraduate intro to neuroscience, it still does not change the fact that inward chloride currents would never depolarize a cardiomyocyte. The sentence should read "...resulted in chloride-driven outward currents that depolarized the cell."

The limitations/application of the of the model remain inadequately addressed in the discussion. The likelihood of using genetically encoded chloride currents treat disorders of cardiac electrical excitability seem low.

Referee #2:

It was a pleasure to see this manuscript back in my inbox, with comments and experimental data addressing points raised at the first review. An obstacle now seems to be the choice between including more information/explanation in this article (that will increase this articles appeal/relevance/understanding) without compromising several ongoing lines of investigation that appear to be well underway by the investigators.

Understandably the authors request a clear dividing line between the scope of this article (predominantly an electrophysiological characterisation of a potential chemigenetic tool-set, where one drug/channel combination emerges more favourably than the other) and future articles they wish to develop which collectively represent the applications.

The main difficulty acquiescing to that request is that the tool-set is so thinly understood/recognized by the community that it is easy to under-appreciate the potential of the work - as is evident in the mixed peer review responses for example.

Unfortunately, although the rebuttal seems to suggest that the comments have all been addressed, in the uploaded version presented by the journals electronic reviewing platform there appear to be a number of omissions, so I wonder if an earlier version has been uploaded by mistake?

Major comments from the first review should not go unanswered.

For example, the response to R2 Q2 suggests some ideas about whether the chemigenetic strategy may be more likely to work in the iPS-CM than the adult-CM would be presented in lines 326-338, but the text is unmodified as far as I can tell; further the rebuttal letter itself does not cover the text that was to be inserted. The request for a reference, or method, for contractile force estimation in EHT does not appear to be present. The source of the small molecules used is now stated, but how they were prepared/used (eg solvent, stock solution, storage) is not.

R2 Q2 on how the tool will perform as a graft matures is not addressed. Based on the rebuttal it would appear fair to at least include a sentence acknowledging that the IPS-CM host changes electrically and mechanically in vivo, and that we don't know how this approach will perform in that transition.

The EHT may be a good model to study that maturation. Currently the rebuttal focusses on the possibility of transgene expression changes during maturation. This was not the point of the original question, which instead was asking about what might happen to this approach when things affecting depolarisation/repolarisation start to change. Can this be reconsidered?

I can accept the position offered by the investigators about not commenting on structural changes in the EHT during treatment and following washout. The appearance of data suggesting diastolic arrest is relevant and welcome, to my eyes it appears that the cells are more relaxed than the normal diastolic state, making me wonder whether there is an increase in super-relaxed myosin perhaps as seen in hibernation? I wish the authors success in exploring the many aspects of this phenomenon, which will need to exclude the known toxicities of prolonged depolarising cardioplegic arrest identified in the clinical world, but that can wait for another day if the editor agrees.

It is a shame that the authors are unable to study voltage or calcium within their EHTs for extended periods of time, these methods are described and could be incorporated into their current experimental strategy, but again will have to be deferred if they can't do that presently.

There are some other minor points related to the revisions that should be possible to address quickly:

1. I am grateful for the extended description of how the clones were produced and characterised. The additional text, and supplementary figure are helpful, but not explicit enough for an audience that may not routinely undertake genome engineering. By way of example, even the rebuttal to the reviewer the reply is a little jumbled - "each clone carrying a single copy per allele" is not the same as "integration of a single copy in the clones used" as the cell is diploid.

It would probably help the audience if it could just be simply stated whether the final clone selected were homozygous, or heterozygous knock-ins.

This is important as expression levels matter for these tools. Typically knock-in approaches give nM range expression in most/all cells, in contrast to viral methods (0.4-4 μ M) which exhibit higher, but variable expression in some, but not all, cells.

2. A related point is the claim about transgene expression - stated to be "high", or "(very) high" in the rebuttal response to R1 Q1b & c. To the authors credit these claims do not make it into the article itself.

Unfortunately, the authors are unable to measure the transgene of interest directly (expression level or subcellular localisation), and only imply expression via the 5' GFP reporter in the construct used, and the response to selective drug by measured current. I wonder, in the absence of formal quantification of PSAM expression (or estimates contained in a back-calculation from channel conductance), if it may be more prudent to describe things a little more cautiously - "transgene expression was present in XX% of cells as suggested by GFP presence" or something to that effect?

As has been established for the optogenetic actuators channel expression level and membrane targeting (both by number of cells, and amount per cell) will determine the feasibility of this approach and perhaps at this stage a different choice of language would be less contentious in the long run?

The authors point out that the detour they took to a stable knock-in strategy may have led to the experimental success - rather than the tempting short cut viral delivery. I think it would be fair for them to emphasise the point in the article (and gain the credit for the up-front effort that may save others the task as the materials increasingly shared within the research community). Viral delivery of these tools is unsatisfactory in this type of model system as it is impossible titrate a drug response if the cellular substrate has variable channel expression - including a potentially large number of cells that express

no transgene. This is a major technical point that is easily overlooked.

3. The answer to R2 Q1c is not satisfactory, specifically the point about why selective inhibition of the graft chemically will not inhibit contractility of the whole heart? I apologise if the question was not clear enough, so forgive me for trying again to get a clear answer (that makes it into the article not just the rebuttal).

As the authors point out - the EHT is an electromechanical syncytium, so whole EHT effects are seen - even if all of the cells in the EHT don't express the transgene that enable chemigenetic control. Will that electrical effect stop at the edge of a graft?

The authors will know that for a decade relatively few cells expressing an optically controlled depolarising actuator can control the whole heart (PMID: 26098449). Relevant to the article under review, the same is true for the optically controlled chloride pump NpHR in zebrafish (PMID: 21071670), where continuous activation of a small number of cells suppresses ventricular activation.

Hence, in the engraftment context - where I quite accept the obvious (that the graft will be selectively targeted by the chemical controlling the transgene) - the question is why won't islands held in a depolarised block (see response to R2 Q3) by this method trigger a depolarised block of the whole heart - with which they are intended to form an electromechanical syncytium?

In other words - why do the authors believe that this approach will allow them to both have the cake and eat it? Is there a suggestion in PMID: 21071670 (see the discussions around movie S6) that the effect the authors want may be possible to obtain?

As application to prevent EA may become a major use of this approach, that the audience could do with being matured for, I would strongly urge the authors to set out why they believe this approach will not cause ventricular arrest, or arrhythmia (R1 final comment) - in theory at least, accepting that practical applications will come in follow-up work.

4. The authors (reply to R2 Q2) posit all EA results from automatic behaviour, and that controlling this alone in the graft will be sufficient to mitigate EA. This is an over-simplification of what is known (for example the authors themselves cite a recent NHP study where little EA was apparent in protocol requiring 20M/60M iPS-CM delivered as 1000-CM spheroids). As the details of EA are not brought out in the current article text there is no need to suggest any adjustments on this point, in this piece, but if the journal wants this technology to be understood in a potential clinical context it may help the readership if this could be addressed in an accompanying editorial to this article, or commissioned review if the editors wished it?

5. The graphical abstract seems a little more cluttered than it needs to be, and from my seat doesn't really summarise the biology as well as it could to the generalist this is targeted at.

Put simply, does the current image convey both the chemical and the genetic basis of the approach clearly? - for me it does not. At the moment the different genetic elements label the base to which the EHT cantilevers are attached rather than the cell. The chemical element looks like a protein. The represented ion channels are different colours not different shapes. Does it really help to have both channels?

Would it be more easily understood if there was a control cell, and a transgene expressing cell? To make space I could remove PSEM89S. I would use the chemical structure of varenicline. In this way I feel the chemigenetic approach is more easily understood.

If the authors want to keep both channel schematics, I would encourage them to address the different elements to make them stand out more clearly, and correct the EC50 for the chemical/genetic combinations tested, they are not identical, but overall I don't think this needs both channels/chemicals represented - this clutters the piece.

6. Figure 3 seems to have adopted a different style (panels are referred to as right or left, and top/bottom, rather than the conventional A, B, C ... adopted elsewhere).

END OF COMMENTS

Reviewing Editor:

Comments for Authors to ensure the paper complies with the Statistics Policy: Please review the journal's statistical policy on reporting the data https://jp.msubmit.net/cgi-bin/main.plex?form_type=display_requirements#statistics This include use of SEM and reporting of P-values.

Comments to the Author:

The manuscript has been re-reviewed by the original referees. Both referees conclude the manuscript will have a moderate to significant impact on the field. However, there are several questions and points that each referee raised that should be addressed. The authors are encouraged to address all these points and ensure the manuscript complies with the Journal's statistical policy.

Senior Editor:

Comments for Authors to ensure the paper complies with the Statistics Policy: See comments from RE

We would like to thank the reviewers for their careful evaluation of our work. We believe that we have addressed all the points raised by the reviewers. For this we included new data in Figure 1C and 7 F and rephrased several paragraphs.

REFEREE COMMENTS

Referee #1:

The authors have made a number of useful additions to the manuscript that improve it greatly the quality of the manuscript. There remain only two issues that should be addressed.

Your sentence, "...resulted in chloride-driven inward currents that depolarized the cell." is incorrect. Although I thoroughly enjoyed the in-depth review of my undergraduate intro to neuroscience, it still does not change the fact that inward chloride currents would never depolarize a cardiomyocyte. The sentence should read "...resulted in chloride-driven outward currents that depolarized the cell."

We believe that the problem here is more linguistic than electrophysiological. The electrophysiological definition of an inward currents is that it depolarizes the cell. This is exactly what we measured in our experiments. An inward current can be cause by cations (as Na^+ , K^+) entering into the cell, but also by anions (as Cl^-) leaving the cell. To omit this confusion, we have now rephrased the sentence and write "...resulted in chloride-driven currents that depolarized the cell."

The limitations/application of the of the model remain inadequately addressed in the discussion. The

likelihood of using genetically encoded chloride currents treat disorders of cardiac electrical excitability seem low.

We believe, that in contrast to other studies with optogenetic tools, our discussion on the in vivo possibilities for the chemogenetic tools was rather modest. However, we now have toned down the respective sentence. It now reads: Additionally, even though far-fetched, PSAM⁴-GlyR could also find application in humans, e.g. in anti-arrhythmic gene therapy or to improve safety of pluripotent stem cell-derived cardiomyocyte transplantation and suppress engraftment-induced arrhythmias which represents a substantial hurdle towards the successful clinical translation of cardiomyocyte transplantation”

Referee #2:

It was a pleasure to see this manuscript back in my inbox, with comments and experimental data addressing points raised at the first review. An obstacle now seems to be the choice between including more information/explanation in this article (that will increase this articles appeal/relevance/understanding) without compromising several ongoing lines of investigation that appear to be well underway by the investigators.

Understandably the authors request a clear dividing line between the scope of this article (predominantly an electrophysiological characterisation of a potential chemigenetic tool-set, where one drug/channel combination emerges more favourably than the other) and future articles they wish to develop which collectively represent the applications.

The main difficulty acquiescing to that request is that the tool-set is so thinly understood/recognized by the community that it is easy to under-appreciate the potential of the work - as is evident in the mixed peer review responses for example.

Unfortunately, although the rebuttal seems to suggest that the comments have all been addressed, in the uploaded version presented by the journals electronic reviewing platform there appear to be a number of omissions, so I wonder if an earlier version has been uploaded by mistake? We appreciate the time and effort the reviewer spent to improve our manuscript, the critical suggestions and the understanding not to present the ongoing studies in which we apply the described tools. We tried to implement some of the ideas that the reviewer mentions in more detail (e.g. lines 419-421). However, while we think that these are fascinating discussions (and actually enjoyed drafting this response) we think that they go far beyond the current manuscript. In our opinion some of the points raised by the reviewer were already discussed in the first version of our manuscript. We have now rephrased the respective paragraphs.

Major comments from the first review should not go unanswered.

For example, the response to R2 Q2 suggests some ideas about whether the chemgenetic strategy may be more likely to work in the iPS-CM than the adult-CM would be presented in lines 326-338, but the text is unmodified as far as I can tell; further the rebuttal letter itself does not cover the text that was to be inserted. The request for a reference, or method, for contractile force estimation in EHT does not

appear to be present. The source of the small molecules used is now stated, but how they were prepared/used (eg solvent, stock solution, storage) is not.

R2 Q2 on how the tool will perform as a graft matures is not addressed. Based on the rebuttal it would appear fair to at least include a sentence acknowledging that the IPS-CM host changes electrically and mechanically *in vivo*, and that we don't know how this approach will perform in that transition.

In our opinion the following sentence in the first submission covered this aspect already: "However, it i) remains open whether the same holds true for adult human CM where cardiac inward rectifying potassium currents (I_{K1}) are much higher (Horváth *et al.*, 2018) and are expected to oppose chloride current-induced depolarization."

We have now expanded this to "hiPSC-CM in the EHT format show signs of maturation, i.e. physiological upstroke velocity and sodium current density but still show signs of immaturity (spontaneous beating). Thus, it remains open whether chloride current-induced depolarization may have the same effect in adult human CM, where cardiac inward rectifying potassium currents (I_{K1}) are much higher than in hiPSC-CM".

To further address this point we included another sentence stating: "The here described tools described could be successfully used to modulate chloride conductivity and contractility in stem-cell derived CM in the EHT format. While these CM are more mature than their 2D-counterparts they are still not fully matured. Whether PSAM-GlyR and PSAM4-GlyR can also be efficient in fully matured CM (or stem cell-derived CM that mature after transplantation) that possess different electrical properties and experience enhanced workload remains to be seen."

Regarding the reference for the contractility measurement. In our opinion Breckwoldt *et al.* (PMID: 28492526) had already covered this aspect. We have now expanded this aspect: "EHTs were measured based on automated video-optical recording. In short, a video camera was placed above the recording chamber, the deflection of the silicone posts was recorded and analyzed with a specific software (CTMV, Pforzheim, Germany). The contraction peaks were analyzed in terms of frequency, force, and contraction (time-to peak, TTP) and relaxation time (RT80) at 80% of peak." We now also reference Mannhardt *et al.* 2016 (PMID: 27211213).

We have included additional information on how the reagents were prepared for the experiments. It now reads: "The compounds varenicline (PZ0004) and picrotoxin (P1675) were provided by Sigma-Aldrich Co. (St. Louis, MO, USA), while the pharmacologically selective effector molecule 89S (PSEM^{89S}, Cat. No. 6426), was obtained from Tocris (Bristol, UK). Varenicline and PSEM^{89S} were prepared as stock solutions in phosphate buffered saline (PBS). Picrotoxin was dissolved in dimethylsulfoxide. The solutions were then aliquoted and stored at -20°C. On the day of the experiment, the stock solutions were warmed to room temperature. Varenicline was diluted from 1 nM to 1 μM, PSEM^{89S} from 1 μM to 1 mM and picrotoxin (PTX) was diluted to 50 mM. Dilutions were prepared using Tyrode's solution for patch clamp and action potential recordings and Ca²⁺ containing DMEM for force measurements."

The EHT may be a good model to study that maturation. Currently the rebuttal focusses on the possibility of transgene expression changes during maturation. This was not the point of the original question, which instead was asking about what might happen to this approach when things affecting depolarisation/repolarisation start to change. Can this be reconsidered?

The experiments were performed after EHT cultivation time of 3 to 5 weeks. At this stage the EHTs still show signs of immaturity, i.e. cell cycle activity 5-10% but also signs of maturation, i.e. physiological upstroke velocity and sodium current density. We have now included data showing that the activation of PSAM-GlyR inhibits contractility even in EHT after a culture period of two months (at this time point EHT show signs of further maturation, i.e. cell cycle activity ~2%), indicating that the system can also work in more matured tissue (new Figure 7F).

I can accept the position offered by the investigators about not commenting on structural changes in the EHT during treatment and following washout. The appearance of data suggesting diastolic arrest is relevant and welcome, to my eyes it appears that the cells are more relaxed than the normal diastolic state, making me wonder whether there is an increase in super-relaxed myosin perhaps as seen in hibernation? I wish the authors success in exploring the many aspects of this phenomenon, which will need to exclude the known toxicities of prolonged depolarising cardioplegic arrest identified in the clinical world, but that can wait for another day if the editor agrees

It is a shame that the authors are unable to study voltage or calcium within their EHTs for extended periods of time, these methods are described and could be incorporated into their current experimental strategy, but again will have to be deferred if they can't do that presently.

We agree that these experiments are of interest and would be a confirmation of a consistent full depolarization block. However, i) we do not have any indication for even a transient interruption of the depolarization block (AP measurements never showed APs upon PSEM⁸⁹⁵ or varenicline exposure). Indeed, when using (sub)threshold varenicline concentrations one could see contractions of individual cardiomyocytes. This effect has never been observed in the here presented data. ii) Even though methods to perform these measurements have been described these are not trivial to implement. However, we now discuss this limitation (" We cannot fully rule out a transient re-activation during prolonged off-switch periods but have no indication for this at present.")

There are some other minor points related to the revisions that should be possible to address quickly:

1. I am grateful for the extended description of how the clones were produced and characterised. The additional text, and supplementary figure are helpful, but not explicit enough for an audience that may not routinely undertake genome engineering. By way of example, even the rebuttal to the reviewer the reply is a little jumbled - "each clone carrying a single copy per allele" is not the same as "integration of a single copy in the clones used" as the cell is diploid. It would probably help the audience if it could just be simply stated whether the final clone selected were homozygous, or heterozygous knock-ins.

We have adjusted this and state that the clones used were homozygous. It now reads: "Homozygous clones were used for this study (clone ASC3 for PSAM⁴-GlyR; Supplementary Figure 1").

This is important as expression levels matter for these tools. Typically knock-in approaches give nM range expression in most/all cells, in contrast to viral methods (0.4-4uM) which exhibit higher, but variable expression in some, but not all, cells.

2. A related point is the claim about transgene expression - stated to be "high", or "(very) high" in the rebuttal response to R1 Q1b & c. To the authors credit these claims do not make it into the article itself.

Unfortunately, the authors are unable to measure the transgene of interest directly (expression level or subcellular localisation), and only imply expression via the 5' GFP reporter in the construct used, and the response to selective drug by measured current. I wonder, in the absence of formal quantification of PSAM expression (or estimates contained in a back-calculation from channel conductance), if it may be more prudent to describe things a little more cautiously - "transgene expression was present in XX% of cells as suggested by GFP presence" or something to that effect?

We have now included an image demonstrating the membranous integration of PSAM-GlyR by alpha-bungarotoxin staining. Alpha-bungarotoxin binds to nicotinic acetylcholine receptors (new Figure 1C). We also agree that it is a good idea to describe things cautiously but consider our wording: "Transgene expression (based on GFP positivity) was $93\pm 7\%$ in hiPSC-CM" and "Transgene expression was $> 90\%$ (indicated by GFP positivity) in iPSC-CM and remained stable over the whole study period" only subtly different to the one the reviewer suggested and no bold over-interpretation.

As has been established for the optogenetic actuators channel expression level and membrane targeting (both by number of cells, and amount per cell) will determine the feasibility of this approach and perhaps at this stage a different choice of language would be less contentious in the long run?

The authors point out that the detour they took to a stable knock-in strategy may have led to the experimental success - rather than the tempting short cut viral delivery. I think it would be fair for them to emphasise the point in the article (and gain the credit for the up-front effort that may save others the task as the materials increasingly shared within the research community). Viral delivery of these tools is unsatisfactory in this type of model system as it is impossible titrate a drug response if the cellular substrate has variable channel expression - including a potentially large number of cells that express no transgene. This is a major technical point that is easily overlooked.

We appreciate the reviewer's comments on this topic. We now emphasize the advantages of a stable knock-in more ("We have chosen a knock-in approach over a viral delivery to achieve a stable transgene expression (per cell but also in the cell population)"). However, while we realize the importance of transgene expression and very much favor our knock-in approach, we also realize that a plethora of studies have used optogenetic tools successfully without ever measuring transgene expression, either with transgenic approaches or by viral delivery. Similar transgene expression has not been studied in the initial description of the chemogenetic tools. Moreover, in our opinion, a transgene expression in the vast majority of the cells can be considered as high or even very high. Nevertheless, we refrained from this interpretation in the original manuscript and used a more cautious description. To further substantiate this point, and not only rely on GFP-expression, we now include a staining with alpha-bungarotoxin that labels PSAM-GlyR demonstrating the membranous localization of PSAM-GlyR (new Figure 1C). Even though quantification is difficult with this strategy it clearly shows that most of the cells express PSAM-GlyR.

3. The answer to R2 Q1c is not satisfactory, specifically the point about why selective inhibition of the graft chemically will not inhibit contractility of the whole heart? I apologise if the question was not clear enough, so forgive me for trying again to get a clear answer (that makes it into the article not just the rebuttal).

As the authors point out - the EHT is an electromechanical syncytium, so whole EHT effects are seen - even if all of the cells in the EHT don't express the transgene that enable chemigenetic control. Will that electrical effect stop at the edge of a graft?

The authors will know that for a decade relatively few cells expressing an optically controlled depolarising actuator can control the whole heart (PMID: 26098449). Relevant to the article under review, the same is true for the optically controlled chloride pump NpHR in zebrafish (PMID: 21071670), where continuous activation of a small number of cells suppresses ventricular activation.

Hence, in the engraftment context - where I quite accept the obvious (that the graft will be selectively targeted by the chemical controlling the transgene) - the question is why won't islands held in a depolarised block (see response to R2 Q3) by this method trigger a depolarised block of the whole heart - with which they are intended to form an electromechanical syncytium?

In other words - why do the authors believe that this approach will allow them to both have the cake and eat it? Is there a suggestion in PMID: 21071670 (see the discussions around movie S6) that the effect the authors want may be possible to obtain?

As application to prevent EA may become a major use of this approach, that the audience could do with being matured for, I would strongly urge the authors to set out why they believe this approach will not cause ventricular arrest, or arrhythmia (R1 final comment) - in theory at least, accepting that practical applications will come in follow-up work.

We realized that we had not described the details of our idea here enough but also that there is some misunderstanding on the reviewer's side with regard to the literature. In PMID: 26098449 an optogenetic actuator was expressed in a few ventricular cells. Similarly, in our recent work Stüdemann et al. Stem Cell Reports 2024 we transplanted cells expressing an optogenetic actuator in the ventricular myocardium. In both cases pulsed photostimulation with frequencies higher than the endogenous rhythm, controlled the heart rate. There is no data on continuous photostimulation in PMID: 26098449 but in our hands continuous photostimulation did not evoke arrhythmias and also did not inhibit contractility of the whole heart. Similar in Stüdemann et al. Circulation 2021 (PMID: 36073365) a depolarization block in a small subset of cardiomyocyte in the ventricle did not induce a depolarization block of the whole heart. In our understanding this is perfectly in line with physiological considerations: by photostimulation (or chemogenetic depolarization) one triggers a single action potential of the engrafted cells (or transduced ventricular CM). This AP can spread to the surrounding myocardium. In case of pulsed depolarization, one triggers repeated APs that will propagate (by activating sodium channels) to the neighboring cells and will be conducted through the whole heart. Situation is at variance in case of continuous depolarization. It is very unlikely that an individual AP, induced at a random time, propagates to the host myocardium because of its long refractory period. Yet, even if it

does it will only cause one premature beat. The shift in diastolic potential will be transmitted through the connexins to the neighboring cells. Because of ohmic conduction, voltage shift drops with larger distance from the cells that actively underwent depolarization. Thus, continuous photostimulation (or activation of chemogenetic tools with drugs) then keeps the engrafted CM in a depolarization state but does not induce APs. Tonic depolarization will be restricted to engrafted CM and (at best) a small rim of neighboring cells. Thus, the majority of surrounding myocardium remains excitable.

We do not agree with the argument we should consider PMID: 21071670. Here prolonged photostimulation of a small number of cells inhibited the whole heart. However, those were highly specialized cells, namely sinoatrial node or atrioventricular node cells. The authors specifically discussed their surprise that when inhibiting the sinoatrial node in the zebrafish larvae no additional pacemaker centers took over and that this stood in contrast to other species. Continuous photostimulation of the ventricular myocardium had no or only minor effects in this model (Figure 1C and D in PMID: 21071670). The reviewer mentions movie S6 of PMID21071670. Here, like the above-mentioned studies, the authors applied a pulsed (they call it periodic) photostimulation. Our idea is to induce a continuous inhibition of the engrafted ventricular CM after transplantation. This will result in one AP (most likely not propagating to the host myocardium and even if only inducing a premature beat), will then keep the engrafted CM in a depolarized state thus renders them in-excitable but will leave the surrounding myocardium excitable. The reviewer mentions the EHT as a syncytium. However, it does not mean that voltage shift in one cell will be transmitted 1:1 to the rest of the heart. In fact, we do have preliminary data in which we switched-off a small subpopulation of the CM in an EHT (by mixing PSAM-GlyR and WT CM). This resulted in drop of function but did not stop the contractility of the whole EHT, further substantiating our hypothesis. In this regard, this does not seem to be an eating the cake and having it situation but whether our idea will be successful to prevent engraftment induced arrhythmias remains to be studied.

We find this topic highly interesting and at least some of the authors share the reviewer's enthusiasm on cardiac regeneration. However, due to space constraints, and as this is point is clearly not the focus of this manuscript, we have oriented ourselves on the work of Wexler et al. 2022 (PMID: 36744466) regarding the discussion on this topic. We have now slightly expanded this paragraph but if the reviewer and the editors think that is worth to elaborate further on this specific point in great detail, we are happy to do so.

We are not sure whether inhibiting EA engraftment arrhythmias will be the main application for these tools but also see a lot of in vitro application possibilities, e.g. we use these tools to assess the correlation between contractility and cell cycle activity and have shared them with several other groups working on a variety of topics.

4. The authors (reply to R2 Q2) posit all EA results from automatic behaviour, and that controlling this alone in the graft will be sufficient to mitigate EA. This is an over-simplification of what is known (for example the authors themselves cite a recent NHP study where little EA was apparent in protocol requiring 20M/60M iPS-CM delivered as 1000-CM spheroids). As the details of EA are not brought out in the current article text there is no need to suggest any adjustments on this point, in this piece, but if the journal wants this technology to be understood in a potential clinical context it may help the readership

if this could be addressed in an accompanying editorial to this article, or commissioned review if the editors wished it?

We respectfully disagree here. At present all large animal studies that have investigated EA point towards automatism as the trigger (e.g. Romagnuolo et al., Stem Cell Reports 2019; Liu et al, Nature Biotech 2019, Marchiano et al. Cell Stem Cell 2023, Selvakumar et al. Nature Cardiovascular Research 2024). While there might be other mechanisms at play, e.g. micro-reentry, we do not consider this interpretation an over-simplification.

5. The graphical abstract seems a little more cluttered than it needs to be, and from my seat doesn't really summarise the biology as well as it could to the generalist this is targeted at.

Put simply, does the current image convey both the chemical and the genetic basis of the approach clearly? - for me it does not. At the moment the different genetic elements label the base to which the EHT cantilevers are attached rather than the cell. The chemical element looks like a protein. The represented ion channels are different colours not different shapes. Does it really help to have both channels?

Would it be more easily understood if there was a control cell, and a transgene expressing cell? To make space I could remove PSEM89S. I would use the chemical structure of varenicline. In this way I feel the chemigenetic approach is more easily understood.

If the authors want to keep both channel schematics, I would encourage them to address the different elements to make them stand out more clearly, and correct the EC50 for the chemical/genetic combinations tested, they are not identical, but overall I don't think this needs both channels/chemicals represented - this clutters the piece.

We have modified the graphical abstract.

6. Figure 3 seems to have adopted a different style (panels are referred to as right or left, and top/bottom, rather than the conventional A, B, C ... adopted elsewhere).

We have adjusted this figure to comply with the overall style.

Dear Dr Weinberger,

Re: JP-RP-2024-286428R2 "Modulation of cardiomyocyte contractility and action potentials with chemogenetic chloride currents" by Muhammed Sönmez, Tim Stüdemann, Christoph Manthey, Anita Covic, Nancy Shehata, Junsoo Im, Barbora Schwarzova, Judith Rössinger, Rajiven Srikantharajah, Aya Shibamiya, Edzard Schwedhelm, Thomas Eschenhagen, Torsten Christ, and Florian Weinberger

Thank you for submitting your revised Research Article to The Journal of Physiology. It has been assessed by the original Reviewing Editor and Referees and has been well received. Some final revisions have been requested.

REVISION CHECKLIST:

We look forward to receiving your revised submission.

Yours sincerely,

Kim Barrett
Senior Editor
The Journal of Physiology

REQUIRED ITEMS FOR REVISION

- Your paper contains Supporting Information of a type that we no longer publish, including supplementary tables and figures. Any information essential to an understanding of the paper must be included as part of the main manuscript and figures. The only Supporting Information that we publish are video and audio, 3D structures, program codes and large data files. Your revised paper will be returned to you if it does not adhere to our Supporting Information Guidelines.

- Papers must comply with the Statistics Policy: https://jp.msubmit.net/cgi-bin/main.plex?form_type=display_requirements#statistics.

In summary:

- If $n \leq 30$, all data points must be plotted in the figure in a way that reveals their range and distribution. A bar graph with data points overlaid, a box and whisker plot or a violin plot (preferably with data points included) are acceptable formats.
- If $n > 30$, then the entire raw dataset must be made available either as supporting information, or hosted on a not-for-profit repository, e.g. FigShare, with access details provided in the manuscript.
- 'n' clearly defined (e.g. x cells from y slices in z animals) in the Methods. Authors should be mindful of pseudoreplication.
- All relevant 'n' values must be clearly stated in the main text, figures and tables.
- The most appropriate summary statistic (e.g. mean or median and standard deviation) must be used. Standard Error of the Mean (SEM) alone is not permitted.
- Exact p values must be stated. Authors must not use 'greater than' or 'less than'. Exact p values must be stated to three significant figures even when 'no statistical significance' is claimed.
- Please include the Abstract Figure Legend text within the main article file.

EDITOR COMMENTS

Reviewing Editor:

Comments for Authors to ensure the paper complies with the Statistics Policy:
The authors do not provide precise P values for statistical tests.

Comments to the Author

:

Both referees agree that the work will have a moderate to significant impact on the field. Please review the journal's statistical policy at https://jp.msubmit.net/cgi-bin/main.plex?form_type=display_requirements#statistics. For a given conclusion to be assessed, the exact p values must be stated to three significant figures (not decimal places) even when 'no statistical significance' is being reported (i.e. for anything >0.001, please report to 3 significant figures, e.g. 0.00236 or 0.523, etc.). These should be stated in the main text, as well as in the figures and their legends and tables. The only exception is if p is less than 0.001, in which case '<' is permitted.

Senior Editor:

Comments for Authors to ensure the paper complies with the Statistics Policy:
See comments from RE

REFEREE COMMENTS

Referee #1:

Thank you for the clarification.

I have no further comments.

Referee #2:

Although I don't agree with all the comments, I think the authors response is fair and considered. I am grateful for the additions that make the piece more accessible. I have no further suggestions and would like to wish the team well for their future work.

END OF COMMENTS

Reviewing Editor:

Comments for Authors to ensure the paper complies with the Statistics Policy:
The authors do not provide precise P values for statistical tests.

Referee #1:

Thank you for the clarification.

I have no further comments.

Referee #2:

Although I don't agree with all the comments, I think the authors' response is fair and considered. I am grateful for the additions that make the piece more accessible. I have no further suggestions and would like to wish the team well for their future work.

We would like to thank the editors and the reviewers again for their careful evaluation of our work. We have now included the P-values in the figures as well as in the manuscript text (lines 190-195, 210-216, 224, 225, 231 and 232)

Dear Associate Professor Weinberger,

Re: JP-RP-2025-286428R3 "Modulation of cardiomyocyte contractility and action potentials with chemogenetic chloride currents" by Muhammed Sönmez, Tim Stüdemann, Christoph Manthey, Anita Covic, Nancy Shehata, Junsoo Im, Barbora Schwarzova, Judith Rössinger, Rajiven Srikantharajah, Aya Shibamiya, Edzard Schwedhelm, Thomas Eschenhagen, Torsten Christ, and Florian Weinberger

We are pleased to tell you that your paper has been accepted for publication in The Journal of Physiology.

- Your paper contains Supporting Information of a type that we no longer publish, including supplementary tables and figures. Any information essential to an understanding of the paper must be included as part of the main manuscript and figures. The only Supporting Information that we publish are video and audio, 3D structures, program codes and large data files. Your revised paper will be returned to you if it does not adhere to our Supporting Information Guidelines.

Yours sincerely,

Kim Barrett
Senior Editor
The Journal of Physiology

If you would like to receive our 'Research Roundup', a monthly newsletter highlighting the cutting-edge research published in The Physiological Society's family of journals (The Journal of Physiology, Experimental Physiology, Physiological Reports, The Journal of Nutritional Physiology and The Journal of Precision Medicine: Health and Disease), please click this link, fill in your name and email address and select 'Research Roundup':

<https://www.physoc.org/journals-and-media/membernews>

- **TRANSPARENT PEER REVIEW POLICY:** To improve the transparency of its peer review process, The Journal of Physiology publishes online as supporting information the peer review history of all articles accepted for publication. Readers will have access to decision letters, including Editors' comments and referee reports, for each version of the manuscript as well as any author responses to peer review comments. Referees can decide whether or not they wish to be named on the peer review history document.
- You can help your research get the attention it deserves! Check out Wiley's free Promotion Guide for best-practice recommendations for promoting your work at: www.wileyauthors.com/eeo/guide. You can learn more about Wiley Editing Services which offers professional video, design, and writing services to create shareable video abstracts, infographics, conference posters, lay summaries, and research news stories for your research at: www.wileyauthors.com/eeo/promotion.
- **IMPORTANT NOTICE ABOUT OPEN ACCESS:** To assist authors whose funding agencies mandate public access to published research findings sooner than 12 months after publication, The Journal of Physiology allows authors to pay an Open Access (OA) fee to have their papers made freely available immediately on publication.

EDITOR COMMENTS

Reviewing Editor:

The authors have modified the manuscript to be under the journal's statistical guidelines for reporting data.